# Can sleep protect memories from catastrophic forgetting?

Oscar C González[1†], Yury Sokolov[1†], Giri P Krishnan[1], Jean Erik Delanois[1,2], Maxim Bazhenov[1]*

[1]Department of Medicine, University of California, San Diego, La Jolla, United States; [2]Department of Computer Science and Engineering, University of California, San Diego, La Jolla, United States

**Abstract** Continual learning remains an unsolved problem in artificial neural networks. The brain has evolved mechanisms to prevent catastrophic forgetting of old knowledge during new training. Building upon data suggesting the importance of sleep in learning and memory, we tested a hypothesis that sleep protects old memories from being forgotten after new learning. In the thalamocortical model, training a new memory interfered with previously learned old memories leading to degradation and forgetting of the old memory traces. Simulating sleep after new learning reversed the damage and enhanced old and new memories. We found that when a new memory competed for previously allocated neuronal/synaptic resources, sleep replay changed the synaptic footprint of the old memory to allow overlapping neuronal populations to store multiple memories. Our study predicts that memory storage is dynamic, and sleep enables continual learning by combining consolidation of new memory traces with reconsolidation of old memory traces to minimize interference.

*For correspondence:
mbazhenov@ucsd.edu

†These authors contributed equally to this work

Competing interests: The authors declare that no competing interests exist.

## Introduction

Animals and humans are capable of continuous, sequential learning. In contrast, modern artificial neural networks suffer from the inability to perform continual learning (*Ratcliff, 1990*; *French, 1999*; *Hassabis et al., 2017*; *Hasselmo, 2017*; *Kirkpatrick et al., 2017*). Training a new task results in interference and catastrophic forgetting of old memories (*Ratcliff, 1990*; *McClelland et al., 1995*; *French, 1999*; *Hasselmo, 2017*). Several attempts have been made to overcome this problem including (a) explicit retraining of all previously learned memories – interleaved training (*Hasselmo, 2017*), (b) using generative models to reactivate previous inputs (*Kemker and Kanan, 2017*), or (c) artificially 'freezing' subsets of synapses important for the old memories (*Kirkpatrick et al., 2017*). These solutions help prevent new memories from interfering with previously stored old memories, however they either require explicit retraining of all past memories using the original data or have limitations on the types of trainable new memories and network architectures (*Kemker and Kanan, 2017*). How biological systems avoid catastrophic forgetting remains to be understood. In this paper, we propose a mechanism for how sleep modifies network synaptic connectivity to minimize interference of competing memory traces enabling continual learning.

Sleep has been suggested to play an important role in learning and memory (*Paller and Voss, 2004*; *Walker and Stickgold, 2004*; *Oudiette et al., 2013*; *Rasch and Born, 2013*; *Stickgold, 2013*; *Weigenand et al., 2016*; *Wei et al., 2018*). Specifically, the role of stage 2 (N2) and stage 3 (N3) of Non-Rapid Eye Movement (NREM) sleep has been shown to help with the consolidation of newly encoded memories (*Paller and Voss, 2004*; *Walker and Stickgold, 2004*; *Rasch and Born, 2013*; *Stickgold, 2013*). The mechanism by which memory consolidation is influenced by sleep is still debated, however, a number of hypotheses have been put forward. Sleep may enable memory consolidation through repeated reactivation or *replay* of specific memory traces during characteristic

sleep rhythms such as spindles and slow oscillations (*Paller and Voss, 2004*; *Clemens et al., 2005*; *Marshall et al., 2006*; *Oudiette et al., 2013*; *Rasch and Born, 2013*; *Weigenand et al., 2016*; *Ladenbauer et al., 2017*; *Wei et al., 2018*; *Xu et al., 2019*). Memory replay during NREM sleep could help strengthen previously stored memories and map memory traces between brain structures. Previous work using electrical (*Marshall et al., 2004*; *Marshall et al., 2006*; *Ladenbauer et al., 2017*) or auditory (*Ngo et al., 2013*) stimulation showed that increasing neocortical oscillations during NREM sleep resulted in improved consolidation of declarative memories. Similarly, spatial memory consolidation has been shown to improve following cued reactivation of memory traces during NREM sleep (*Paller and Voss, 2004*; *Oudiette et al., 2013*; *Oudiette and Paller, 2013*; *Papalambros et al., 2017*). Our recent computational studies found that sleep dynamics can lead to replay and strengthening of recently learned memory traces (*Wei et al., 2016*; *Wei et al., 2018*; *Wei et al., 2020*). These studies point to the critical role of sleep in memory consolidation.

Can neuroscience inspired ideas help solve the catastrophic forgetting problem in artificial neuronal networks? The most common machine learning training algorithm – backpropagation (*Rumelhart et al., 1986*; *Werbos, 1990*; *Kriegeskorte, 2015*) – is very different from plasticity rules utilized by brain networks. Nevertheless, we have recently seen a number of successful attempts to implement high level principles of biological learning in artificial network designs, including implementation of the ideas from 'Complementary Learning System Theory' (*McClelland et al., 1995*), according to which the hippocampus is responsible for the fast acquisition of new information, while the neocortex would more gradually learn a generalized and distributed representation. These ideas led to interesting attempts of solving the catastrophic forgetting problem in artificial neural networks (*Kemker and Kanan, 2017*). While few attempts have been made to implement sleep in artificial networks, one study suggested that sleep-like activity can increase storage capacity in artificial networks (*Fachechi et al., 2019*). We recently found that implementation of a sleep-like phase in artificial networks trained using backpropagation can dramatically reduce catastrophic forgetting, as well as improve generalization performance and transfer of knowledge (*Krishnan et al., 2019*; *Tadros et al., 2020*). However, despite this progress, we are still lacking a basic understanding of the mechanisms by which sleep replay affects memories, especially when new learning interferes with old knowledge.

The ability to store and retrieve sequentially related information is arguably the foundation of intelligent behavior. It allows us to predict the outcomes of sensory situations, to achieve goals by generating sequences of motor actions, to 'mentally' explore the possible outcomes of different navigational or motor choices, and ultimately to communicate through complex verbal sequences generated by flexibly chaining simpler elemental sequences learned in childhood. In our new study, we trained a network, capable of transitioning between sleep-like and wake-like states, to learn spike sequences in order to identify mechanisms by which sleep allows consolidation of newly encoded memory sequences and prevents damage to old memories. Our study predicts that during a period of sleep, following training of a new memory sequence in awake, both old and new memory traces are spontaneously replayed, preventing forgetting and increasing recall performance. We found that sleep replay results in fine tuning of the synaptic connectivity matrix encoding the interfering memory sequences to allow overlapping populations of neurons to store multiple competing memories.

## Results

The network model, used in our study, represents a minimal thalamocortical architecture implementing one cortical layer (consisting of excitatory pyramidal (PY) and inhibitory (IN) neurons) and one thalamic layer (consisting of excitatory thalamic relay (TC) and inhibitory reticular thalamic (RE) neurons) – with all neurons simulated by Hodgkin-Huxley models (*Figure 1A*). These models were built upon neuron models we used in our earlier work (*Krishnan et al., 2016*; *Wei et al., 2016*; *Wei et al., 2018*). This model exhibits two primary dynamical states of the thalamocortical system – awake, characterized by random asynchronous firing of all cortical neurons, and slow-wave sleep (SWS), characterized by slow (<1 Hz) oscillations between Up (active) and Down (silent) states (*Blake and Gerard, 1937*; *Steriade et al., 1993*; *Steriade et al., 2001*). Transitions between sleep and awake (*Figure 1B/C*) were simulated by changing network parameters to model effect of neuromodulators (*Krishnan et al., 2016*). While the thalamic population was part of the network, its role

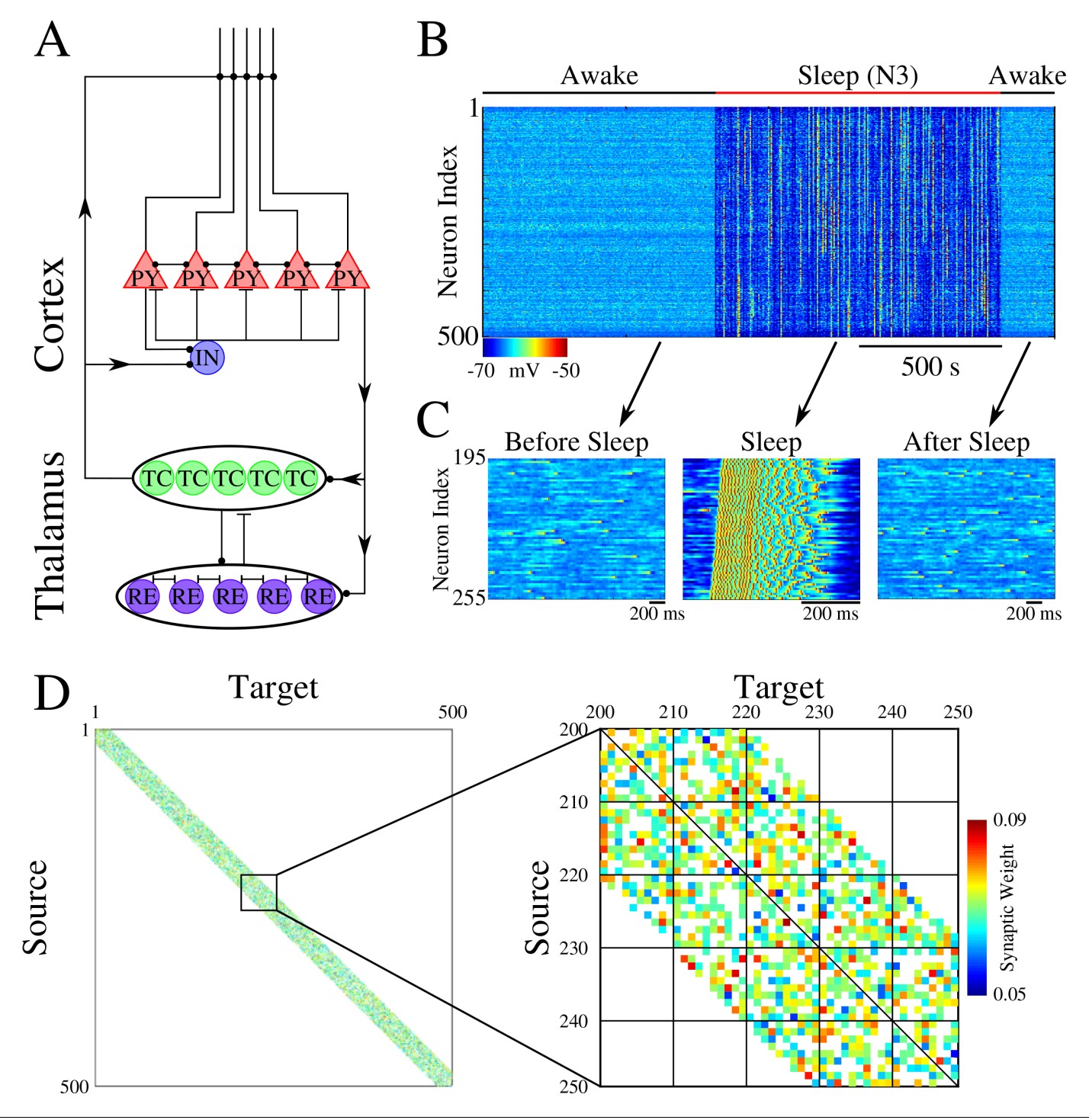

**Figure 1.** Network architecture and baseline dynamics. (A) Basic network architecture (PY: excitatory pyramidal neurons; IN: inhibitory interneurons; TC: excitatory thalamocortical neurons; RE: inhibitory thalamic reticular neurons). Excitatory synapses are represented by lines terminating in a dot, while inhibitory synapses are represented by lines terminating in bars. Arrows indicate the direction of the connection. (B) Behavior of a control network exhibiting wake-sleep transitions. Cortical PY neurons are shown. Color represents the voltage of a neuron at a given time during the simulation (dark blue – hyperpolarized potential; light blue / yellow – depolarized potential; red - spike). (C) Zoom-in of a subset of neurons from the network in B (time is indicated by arrows). Left and right panels show spontaneous activity during awake-like state before and after sleep, respectively. Middle panel shows example of activity during sleep. (D) Left panel shows the initial weighted adjacency matrix for the network in B. The color in this plot represents the strength of the AMPA connections between PY neurons, with white indicating the lack of synaptic connection. Right panel shows the initial weighted adjacency matrix for the subregion indicated on the left.

was limited to help simulate realistic Up and Down state activity (*Bazhenov et al., 2002*), as all synaptic changes occurred in the cortical population. The initial strength of the synaptic connections between cortical PY neurons was Gaussian distributed (*Figure 1D*).

We set probabilistic connectivity (p=0.6) between excitatory cortical neurons within a defined radius ($R_{AMPA(PY-PY)}$=20). Only cortical PY-PY connections were plastic and regulated by spike-timing dependent plasticity (STDP). During initial training, STDP was biased for potentiation to simulate elevated levels of acetylcholine (*Blokland, 1995*; *Shinoe et al., 2005*; *Sugisaki et al., 2016*). During testing/retrieval, STDP was balanced (LTD/LTP = 1). STDP remained balanced during both sleep and interleaved training (except for few selected simulations where we tested effect of unbalancing STDP) to allow side by side comparisons. For details, please see *Methods and Materials*.

Temporally structured sequences of events are a common type of information we learn, and they are believed to be represented in the brain by sequences of neuronal firing. Therefore, in this study we represent each memory pattern as an ordered sequence, S, of activations of populations of cortical neurons (e.g., A→B→. . .), where each 'letter' (e.g., A) labels a population of neurons, so each memory could be labeled by a unique 'word' of such 'letters'. We considered memory patterns represented by non-overlapping populations of neurons as well as memory patterns sharing neurons but with a different activation order, for example, A→B→C vs. C→B→A. This setup can mimic, for example, *in vivo* experiments with a rat learning a track, including: (a) running in one direction on a linear track (*Mehta et al., 1997*) would be equivalent to a sequence training ('A→B→C', 'A→B→C',. . .); (b) forwards and backwards running on a linear track (*Navratilova et al., 2012*) would be equivalent to interleaved sequences training ('A→B→C', 'C→B→A', 'A→B→C',. . .); (c) running on a belt track first only in one direction and then in reverse one (e.g., using Virtual Reality (VR) apparatus) would be equivalent to first learning a sequence ('A→B→C', 'A→B→C',. . .) and then the opposite one ('C→B→A', 'C→B→A',. . .).

In our model, training always occurred in the awake state and no input was delivered to the network in the sleep state. Testing was also done in the awake state; during test sessions, the model was only presented with input to the first group (e.g., A) to test for pattern completion for the trained sequence (e.g., A→B→C→. . .). Performance was calculated based on the distance between the trained pattern (template) and the response during testing. The awake state included multiple testing sessions: before training, after training/before sleep, and after sleep. For details, please see *Methods and Materials*.

The paper is organized as follows. We first consider the scenario of two memory sequences trained at different (non-overlapping) network locations. We show that SWS-like activity after training leads to sequence replay, synaptic weight changes, and performance increases during testing after sleep. Next, we focus on the case of two sequences trained in opposite directions over the same population of neurons. We show that in such a case training a new sequence in awake would 'erase' an old memory. However, if a sleep phase is implemented before complete destruction of the old memory, both memory sequences are spontaneously replayed during sleep. As a result of replay, each sequence allocates its own subset of neurons/synapses, and performance increases for both sequences during testing after sleep. We complete the study with a detailed analysis of synaptic weight changes and replay dynamics during the sleep state to identify mechanisms of memory consolidation and performance increase. In supplementary figures, we compare sleep replay with interleaved training and show that sleep achieves similar or better performance but without explicit access to the training data.

## Training of spatially separated memory sequences does not lead to interference

First, we trained two memory patterns, S1 and S2, sequentially (first S1 and then S2) in spatially distinct regions of the network as shown in *Figure 2A*. Each memory sequence was represented by the spatio-temporal pattern of 5 sequentially activated groups of 10 neurons per group. A 5 ms delay was included between stimulations of subsequent groups within a sequence. S1 was trained in the population of cortical neurons 200–249 (*Figure 2B*, top). Training S1 resulted in an increase of synaptic weights between participating neurons (*Figure 2D*, left) and an increase in performance on sequence completion (*Figure 2B/C*, top). When the strength of the synapses in the direction of S1 increased, synapses in the opposite direction showed a reduction consistent with the STDP rule (see *Methods and Materials*). The second sequence, S2, was trained for an equal amount of time as S1

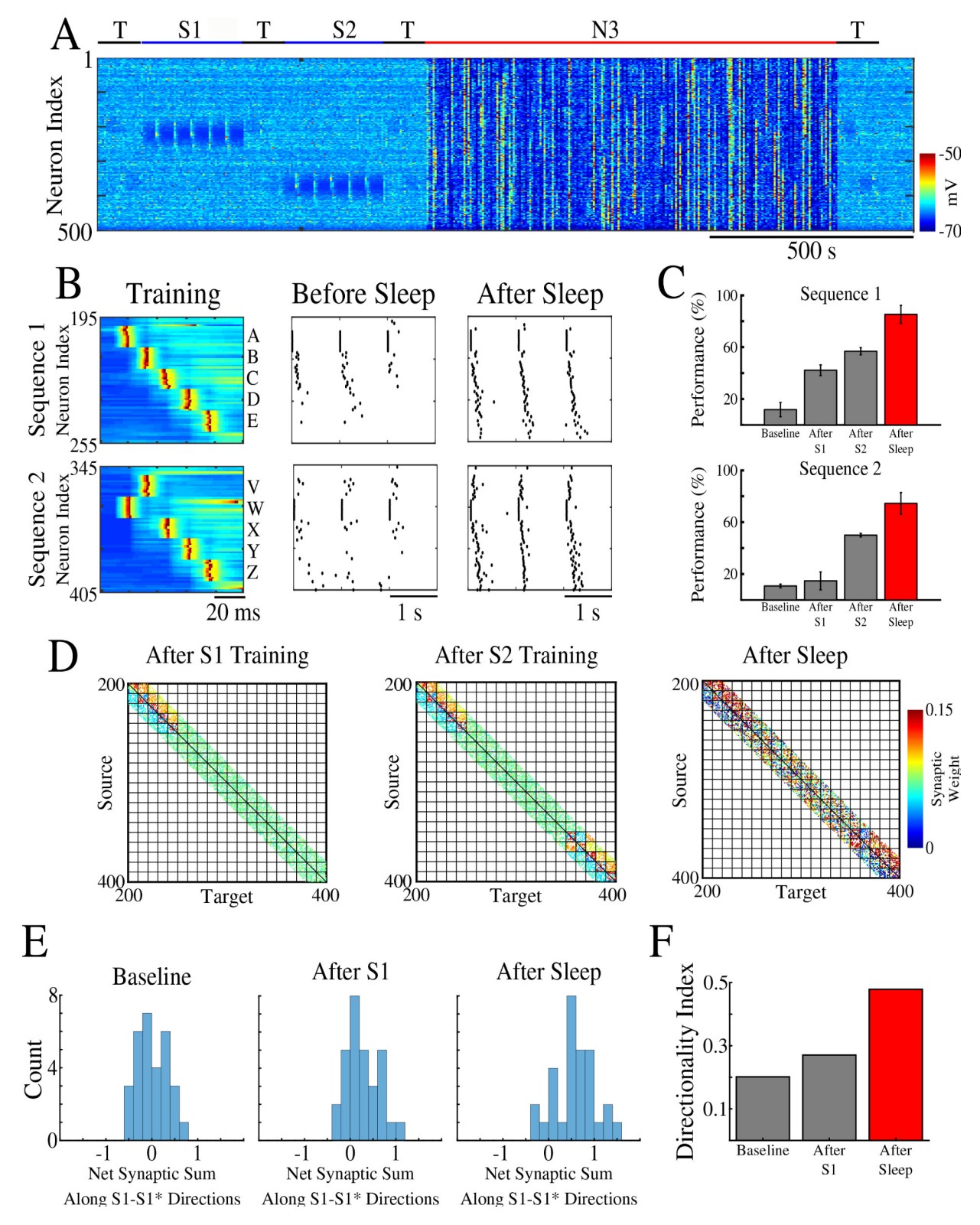

**Figure 2.** Two spatially separated memory sequences show no interference during training and both are strengthened by subsequent sleep. (A) Network activity during periods of testing (T), training of two spatially separated memory sequences (S1/S2), and sleep (N3). Cortical PY neurons are shown. Color indicates voltage of neurons at a given time. (B) Left panels show an example of training sequence 1 (S1, top) and sequence 2 (S2, bottom). Middle panels show examples of testing both sequences prior to sleep. Right panels show examples of testing after sleep. Note, after sleep,
*Figure 2 continued on next page*

*Figure 2 continued*

both sequences show better completion. (C) Performance of S1 and S2 completion before any training (baseline), after S1 training, after S2 training, and after sleep (red). (D) Synaptic weight matrices show changes of synaptic weights in the regions trained for S1 and S2. Left panel shows weights after training S1; middle panel shows weights after training S2; right panel shows weights after sleep. Color indicates strength of AMPA synaptic connections. (E) Distributions of the net sum of synaptic weights each neuron receives from all the neurons belonging to its left neighboring group (S1 direction) vs its right neighboring group (opposite direction, defined as S1* direction below) within a trained region at baseline (left), after S1 training (middle) and after sleep (right). (F) Synaptic weight-based directionality index before/after training (gray bars) and after sleep (red bar).

The online version of this article includes the following figure supplement(s) for figure 2:

**Figure supplement 1.** Sleep replay improves performance for complex non-linear sequences.

but in a different population of neurons 350–399 (W-V-X-Y-Z, *Figure 2B*, bottom). Training of S2 also resulted in synaptic weight changes (*Figure 2D*, middle) and improvement in performance (*Figure 2B/C*, bottom). Importantly, training of S2 did not interfere with the weight changes encoding S1 because both sequences involved spatially distinct populations of neurons (compare *Figure 2D*, left and middle). It should be noted that though testing resulted in reactivation of memory traces, there was little change in synaptic weights during testing periods because of a relatively small number of pre/post spike events. (Simulations where STDP was explicitly turned off during all testing periods exhibited similar results to those presented here.)

We next calculated the net sum of synaptic weights each neuron received from all neurons belonging to its left vs right neighboring populations (e.g., total input to a neuron $B_i$, belonging to group B, that it received from all the neurons in group A vs all the neurons in group C) and we analyzed the difference of these net weights. The initial distribution was symmetric reflecting the initial state of the network (*Figure 2E*, left). After training, it became asymmetric, indicating stronger input from the left groups (i.e., total input to $B_i$ from all the neurons in group A was larger than that from all the neurons in group C) (*Figure 2E*, middle). These results are consistent with *in vivo* recordings from a rat running in one direction on a linear track (*Mehta et al., 1997*), where this phenomenon was called 'receptive field backwards expansion', i.e., neurons representing locations along the track became asymmetrically coupled such that activity in one group of neurons (one location) led to activation of the next group of neurons (new location) even before the corresponding input occurred (before the animal moved to the new location).

After successful training of both sequences, the network went through a period of sleep (N3 in *Figure 2A*) when no stimulation was applied. After sleep, synaptic weights for both memory sequences revealed strong increases in the direction of their respective activation patterns and further decreases in the opposing directions (*Figure 2D*, right). In line with our previous work (*Wei et al., 2018*), these changes were a result of sequence replay during the Up states of slow oscillation (see next section for details). Synaptic strengthening increased the performance on sequence completion after sleep (*Figure 2B*, right; 2C, red bar). Analysis of the net synaptic input to each neuron from its left vs right neighboring groups, revealed further shift of the synaptic weight distribution (*Figure 2E*, right). This predicts that SWS following linear track training would lead to further receptive field backwards expansion in the cortical neurons. To quantify this asymmetry we calculated a 'directionality index', *I*, for synaptic weights (similar to *Navratilova et al., 2012* but using synaptic weights), based on synaptic input to each neuron from its left vs right neighboring populations ('*Directionality Index*'=0 if all the neurons receive the same input from its left vs right neighboring groups and '*Directionality Index*'=1 if all the neurons receive input from one 'side' only; see *Methods and Materials* for details). This analysis showed an increase in the directionality index from naive to trained cortical networks and further increase after sleep (*Figure 2F*). Note, that the backwards expansion of the place fields was reset between sessions in CA1 (*Mehta et al., 1997*), but not in CA3 (*Roth et al., 2012*), where the backward shift gradually diminished across days, possibly as memories became hippocampus independent (see *Discussion*).

The goal of this study was to reveal basic mechanisms of replay and therefore we focus on the 'simple' linear (e.g., S1) memory sequences. Our results, however, can be generalized to much more complex non-linear sequences (see *Figure 2—figure supplement 1*). In simulations from *Figure 2—figure supplement 1*, training a sequence in awake was not long enough to ensure reliable pattern completion, however, performance was significantly improved after replay during SWS.

## Sleep replay improves pattern completion performance for memory sequences

Why do SWS dynamics lead to improvement in memory performance? The hypothesis is that memory patterns trained in awake are spontaneously replayed during sleep. With this in mind, we next analyzed the network firing patterns during Up states of the slow oscillation to identify replay. We focused our analysis on pairs of neurons (as opposed to the longer sequences) because (a) having different elementary units of a sequence (neuronal pairs) replayed independently would still be sufficient to strengthen the entire sequence; (b) *in vivo* data suggest that memory sequence replay often involves random subsets of the entire sequence (e.g., *Euston et al., 2007*; *Roumis and Frank, 2015*; *Joo and Frank, 2018*; *Swanson et al., 2020*); (c) we want to compare results in this section to the analysis of the overlapping opposite sequences in the following sections, however, we could not reliably detect replay of the full sequences in the latter case possibly because of highly overlapping spiking between sequences.

For each synapse in direction S1 (we refer to it below as S1 synapse) and each Up state, we (a) calculated the time delay between nearest pre/post spikes; (b) transformed this time delay through an STDP-like function to obtain a value characterizing its effect on synaptic weight; and (c) calculated the total net effect of all such spike events. This gave us a net weight change for a given synapse during a given Up state. If we observed a net weight increase, we labeled this S1 synapse as being preferentially replayed during a given Up state. Finally, we counted all the Up states where a given synapse was replayed as defined above. This procedure is similar to off-line STDP, however, instead of weight change over entire sleep, we obtained the number of Up states where a synapse in the direction of S1 was (preferentially) replayed.

*Figure 3A* shows, for each synapse in the direction of S1, the total change of its synaptic strength across entire sleep (Y-axis) vs number of Up states when that synapse was replayed (X-axis). As expected, it shows a strong positive correlation. Synaptic weight changes became negative when the number of Up states where an S1 synapse was replayed dropped below half of the total number of Up states (blue vertical line in *Figure 3A*). In *Figure 3B* we plotted only those S1 synapses which were replayed reliably – for more that 66% of all Up states (dotted line in *Figure 3A*). We found such synapses between all neuronal groups (gray boxes in *Figure 3B*) as well as between neurons within groups.

In *Figure 3C*, we illustrated all the synapses identified in the analysis in *Figure 3B*, that is, synapses that were replayed reliably (in more than 66% of all Up states) in direction of S1. We also colored in blue neurons receiving at least one of these synapses as identified in *Figure 3B*. We concluded that there were multiple direct and indirect synaptic pathways connecting the first (A) and last (E) groups of neurons that were replayed reliably during sleep. These synapses increased their strength which explains reliable memory recall during testing after sleep.

## Sequential training of overlapping memory sequences results in interference

We next tested whether our network model shows interference during awake when a new sequence (S1*) (*Figure 4A*) is trained in the same population of neurons as the earlier old sequence (S1). S1* included the same exact groups of neurons as S1, but the order of activation was reversed, that is, the stimulation order was E-D-C-B-A (*Figure 4B*). S2 was once again trained in a spatially distinct region of the network (*Figure 4A/B*). Testing for sequence completion was performed immediately after each training period. This protocol can represent two somewhat different training scenarios: (a) two competing memory traces (S1 and S1*) are trained sequentially before sleep; (b) the first (old) memory S1 is trained and then consolidated during sleep followed by training of the second (new) memory S1* followed by another episode of sleep. We explicitly tested both scenarios and they behaved similarly, so in the following we discuss the simpler case of two sequentially trained memories followed by sleep. This setup can simulate *in vivo* experiments with a rat running on a belt in a VR apparatus, first in one direction only (learning S1) and then in the opposite direction (learning S1*). An example of the second scenario is presented in *Figure 5—figure supplement 1* and discussed below.

In the model, training S1 increased performance of S1 completion (*Figure 4C*, top/left). It also led to decrease in performance for S1* below its baseline level in the 'naive' network (*Figure 4C*,

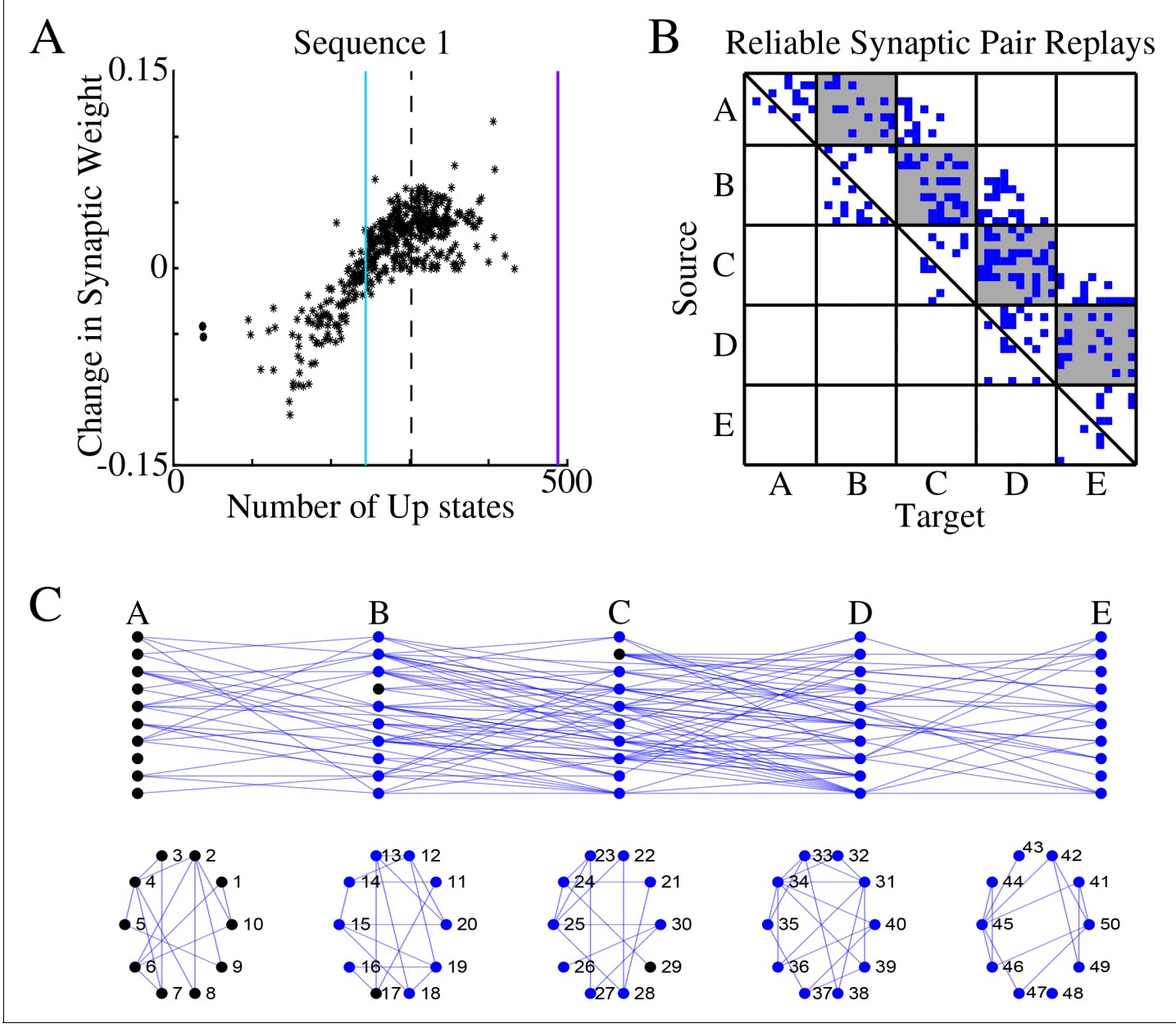

**Figure 3.** Sleep replay strengthens synapses to improve memory recall. (**A**) Change in synaptic weight over entire sleep period as a function of the number of Up states where a given synapse was replayed. Each star represents a synapse in the direction of S1. Dashed line indicates the threshold (66% of Up states) used to identify synapses that are replayed reliably for analysis in B; purple line indicates the maximum number of Up states; blue line demarcates the 50% mark of the total number of Up states. (**B**) Thresholded connectivity matrix indicating synaptic connections (blue) showing reliable replays in the trained region. Grey boxes highlight between group connections. (**C**) Network's graph showing between group (top) and within group (bottom) connections. Edges shown here are those synapses which revealed reliable replays of S1 as shown in B. Nodes are colored blue if they receive at least one of the synapses identified in panel B.

bottom/left). (Note that even a naive network displayed some above zero probability to complete a sequence depending on the initial strength of synapses and spontaneous network activity). Training S2 led to an increase in S2 performance (S1 performance also increased, most-likely due to the random reactivation of S1 in awake). Subsequent training of S1* resulted in both a significant increase in S1* performance and a significant reduction of S1 performance (*Figure 4C*). To evaluate the impact of S1* training on S1 performance, we varied the duration of S1* (later memory) training (*Figure 4D*). Increasing the duration of S1* training correlated with a reduction of S1 performance

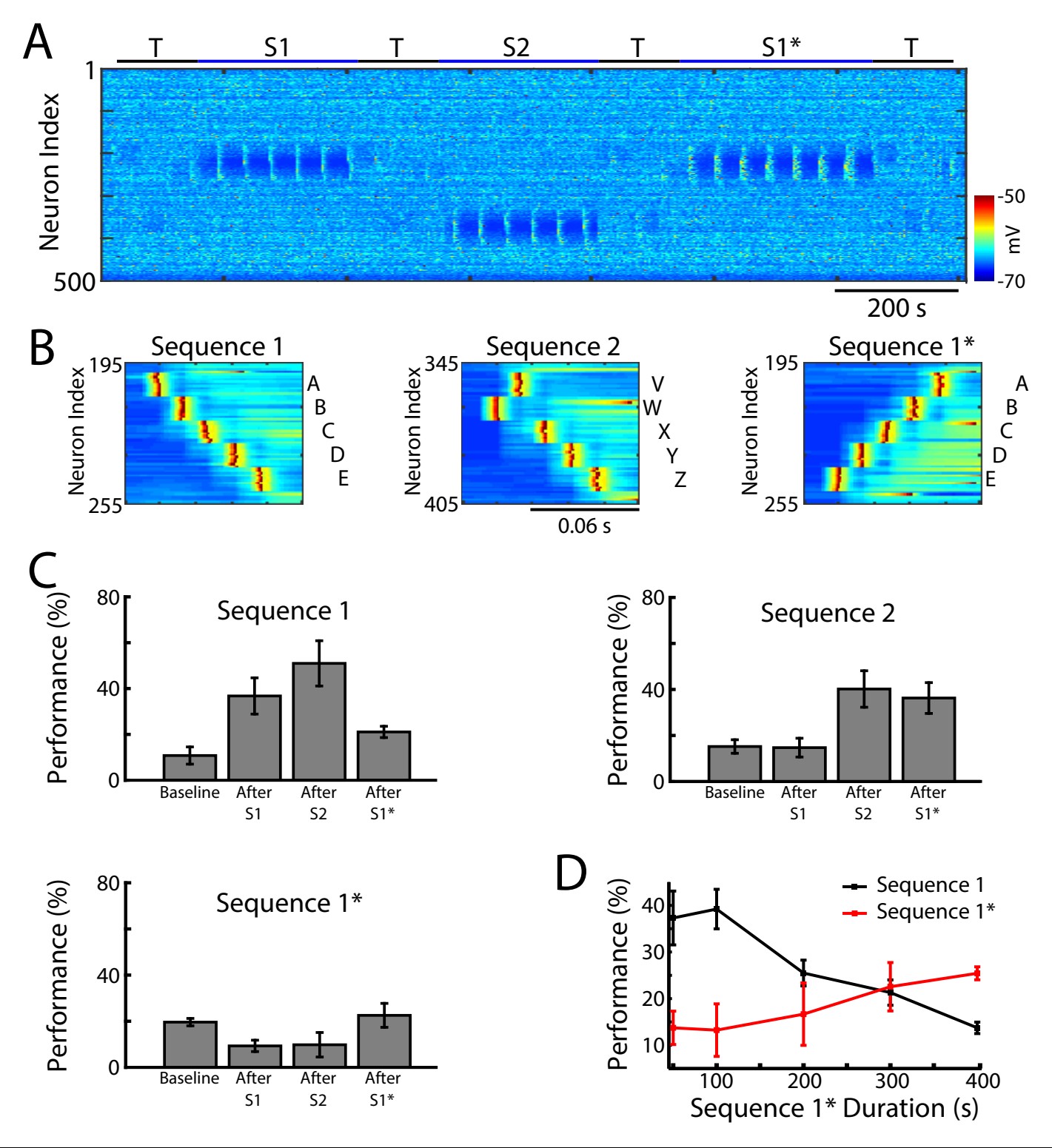

**Figure 4.** Training of overlapping memory sequences results in catastrophic interference. (A) Network activity (PY neurons) during training and testing periods for three memory sequences in awake-like state. Note, sequence 1 (S1) and sequence 1* (S1*) are trained over the same population of neurons. Color indicates the voltage of the neurons at a given time. (B) Examples of sequence training protocol for S1 (left), S2 (middle), and S1* (right). (C) Performances for the three sequences at baseline, and after S1, S2 and S1* training. Training of S1* leads to reduction of S1 performance. (D) Performance of S1 (black) and S1* (red) as a function of S1* training duration. Note that longer S1* training increases degradation of S1 performance. *Figure 4 continued on next page*

*Figure 4 continued*

The online version of this article includes the following figure supplement(s) for figure 4:

**Figure supplement 1.** Interleaved training of the old and new memory sequences prevents the old sequence from forgetting and improves performance for both memories.

up to the point when S1 performance was reduced to its baseline level (*Figures 4D* and 400 sec training duration of S1\*). This suggests that sequential training of two memories competing for the same population of neurons results in memory interference and catastrophic forgetting of the earlier memory sequence.

The model predicts that in experiments with a rat running on a belt in a VR apparatus, training the backward direction after training the forward one would 'erase' the effect of the forward training. While we are not aware of such experiments, studies done with a rat running forward and backward on a liner track (*Navratilova et al., 2012*), which would be equivalent to interleaved training S1→ S1\*→ S1→ S1\*…., revealed that, in the hippocampus, spatial sequences of opposite direction are rapidly orthogonalized, largely on the basis of differential head direction system input, to accommodate both trainings. Thus, at each location, some neurons had their receptive field expanded in one direction and others in the opposite direction (*Navratilova et al., 2012*). To compare our model with these data, we tested interleaved training of S1 and S1\* (*Figure 4—figure supplement 1*) and found performance increase for both sequences. Importantly, in agreement with *in vivo* data, different neurons became specific for S1 vs S1\* as reflected in the overall increase of the directionality index (*Figure 4—figure supplement 1F*). In the next section we test if sleep can achieve the same goal.

## Sleep prevents interference and leads to performance improvement for overlapping memories

So far we found that when a single sequence was trained, it replayed spontaneously during sleep resulting in improvement in performance (*Figures 2* and *3*). For two opposite sequences trained in the same network location we found competition and interference during sequential training in awake (*Figure 4*). However, when the same two sequences were trained using alternating protocol (interleaved training), both increased in performance (*Figure 4—figure supplement 1C*). We next tested the effect of SWS following sequential training of two opposite sequences in awake. Two outcomes are possible: (a) the stronger sequence could dominate replay and eventually suppress the weaker one, or (b) both sequences can be replayed during sleep and increase in performance after sleep. To test these possibilities, we simulated SWS (N3) after the sequences S1/S2/S1\* were trained sequentially in the awake state (S1→ S1→…→ S2→ S2→…→ S1\*→ S1\*→…) (*Figure 5A*), as described in the previous sections (*Figures 2* and *4*). We stopped training the new memory S1\* before the old memory trace S1 was completely erased (300 sec of S1\* training, see *Figure 4D*). Since we biased STDP towards LTP during awake, both memories S1 and S1\* showed above baseline performance after training.

We found that sleep improves sequence completion performance for all three memories, including competing memory traces – S1 and S1\*. *Figure 5B* shows raster plots of the spiking activity before vs after sleep, which revealed significant improvements in sequence completion. These results are summarized in (*Figure 5C*). Thus, we predict that sleep replay is not only able to reverse the damage caused to the old memory (S1) following S1\* training, but it can enhance S1 performance at the same time as it enhances performance of S1\*.

As for a single sequence, we next calculated the net sum of synaptic weights each neuron received from all the neurons belonging to its left vs right neighboring groups, and we analyzed the difference of these net weights. The initial distribution was symmetric reflecting the initial state of the network (*Figure 5D*, left). After S1 training, the distribution became asymmetric, indicating stronger input from the left (*Figure 5D*, middle/left). Training the opposite sequence, S1\*, reversed the process and the distribution became more symmetric again, however, it also became wider with some neurons in each population preferring sequence S1 (i.e., for some group B neurons, $B_i$, input from group A was stronger than input from group C) and others preferring S1\* (i.e., for other group B neurons, $B_j$, input from group C was stronger than input from group A) (*Figure 5D*, middle/right).

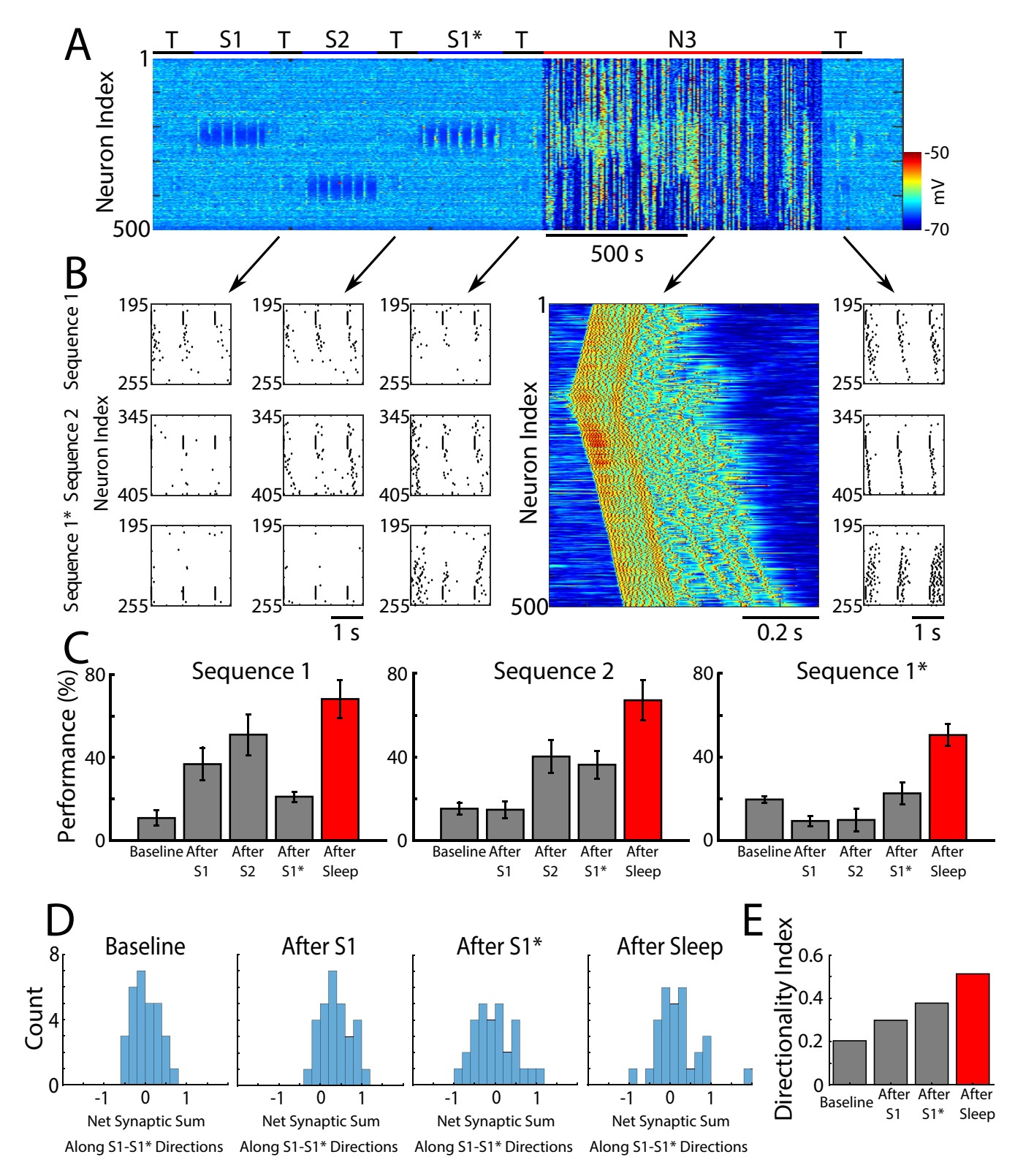

**Figure 5.** Sleep prevents the old memory sequence from forgetting and improves performance for all memories. (**A**) Network activity (PY neurons) during sequential training of sequences S1/S2/S1* (blue bars) followed by N3 sleep (red bar). No stimulation was applied during sleep. (**B**) Examples of testing for each trained memory at different times. The top row corresponds to the testing of S1, middle is testing of S2, and bottom is testing of S1*. Heatmap shows characteristic cortical Up state during SWS. (**C**) Testing of S1, S2, and S1* shows damage to S1 after training S1*, and increase in

*Figure 5 continued on next page*

*Figure 5 continued*

performance for all three sequences after sleep (red bars). (**D**) Distributions of the net sum of synaptic weights each neuron receives from all the neurons belonging to its left vs right neighboring groups within a trained region at baseline (left), after training S1 (middle/left), after training S1* (middle/right), and after sleep (right). Wider distribution indicates presence of neurons that are strongly biased to one sequence or the other. (**E**) Synaptic weight-based directionality index before/after training (gray bars) and after sleep (red bar).

The online version of this article includes the following figure supplement(s) for figure 5:

**Figure supplement 1.** Training of a new memory that interferes with previously consolidated old memory leads to forgetting that can be reversed by subsequent sleep.

After SWS, the width of the distribution further increased indicating that sleep, similar to interleaved training, changes the network connectivity to develop neurons which become strongly specific for one sequence or another (*Figure 5D*, right). The synaptic weight-based directionality index that summarizes these changes (see above and *Methods and Materials* for details) also increased after sleep (*Figure 5E*).

Our study predicts that in experiments with a rat running on a belt in a VR apparatus, training the backward direction after training the forward one can damage (erase) the effect of forward training, however, SWS following training can reverse the damage. Additionally, similar to interleaved training (*Navratilova et al., 2012*), directionality index should increase after SWS.

As we mentioned previously, the training protocol we have focused on in this study was of two memories trained sequentially before sleep. We have also tested the scenario where the first (old) memory is trained and consolidated during sleep before the second (new) memory is trained and then consolidated during a second period of sleep (*Figure 5—figure supplement 1*). The main results from both training protocols remain the same. Thus, performance for S1 improved after first episode of sleep (initial consolidation) (*Figure 5—figure supplement 1B,C*). Training new memory S1* in the same population of neurons damaged S1 and led to improvement of S1*. Consistent with empirical results on proactive interference (*McDevitt et al., 2015*), training S1* took longer in that scenario to achieve a high level of performance. Note, that even longer training of S1* further improved its performance but could also completely erase S1 (*Figure 5—figure supplement 1D*). Finally, both S1 and S1* showed an improvement after a subsequent episode of sleep (*Figure 5—figure supplement 1B,C*). Thus, the training paradigm 'S1→ sleep→ S1*→ sleep' shows qualitatively similar results to the 'S1→ S1*→ sleep' paradigm. This result is also consistent with the 'Complementary Learning Systems Theory' prediction that the old memories interfering with new learning have to be replayed during new phase of memory consolidation to avoid forgetting (*McClelland et al., 2020*).

## Competing memories are replayed spontaneously during Up states of slow oscillation

In this section we focus our analysis on the competing sequences S1 and S1*. We asked the following questions: (a) What kind of network dynamics during Up states of SWS allows for replay and improvement of both memory traces S1 and S1*? (b) Do the same neurons participate in replay of both sequences or do different subsets of neurons uniquely represent each memory? (c) Do both memory sequences replay during the same Up state or do different Up states become biased for replay of one memory or the other?

We performed spike timing analysis similar to what we did for S1 alone (*Figure 3*), but we now analyzed separately synaptic connections in direction of S1 and S1*. *Figure 6A* plots, for each synapse in direction of S1 (left) and S1* (right), the net change in synaptic strength across the entire sleep period vs total number of Up states (slow-waves) where that synapse was preferentially replayed. As before, we found a strong positive correlation. We next plotted only those synapses which replayed reliably – more that 66% of all Up states (*Figure 6B*). We found that such synapses exist between all neuronal groups and for both sequences (in *Figure 6B* blue color indicates synapses in the direction of S1 and red in the direction of S1*). This analysis revealed two important properties. First, after sleep, each pair of neurons preferentially supported only one sequence, S1 or S1* (note that the connectivity matrix in *Figure 6B* is strictly asymmetric). Second, individual neurons can be divided into two groups - those participating reliably in only one sequence replay (either S1 or

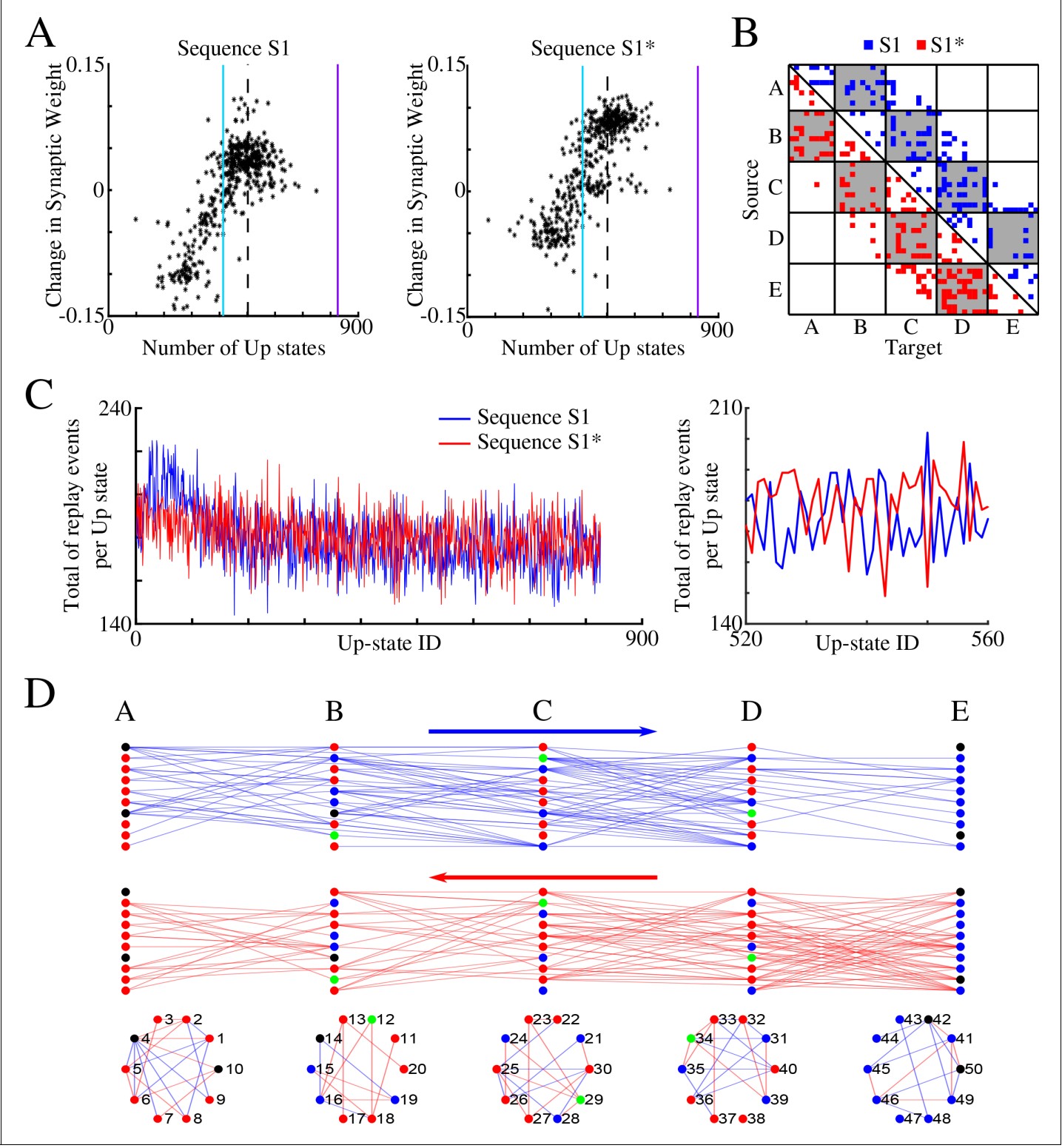

**Figure 6.** Sleep promotes replay of both overlapping memory sequences during each Up state. (**A**) Change in synaptic weight over entire sleep period as a function of the number of Up states where a given synapse was preferentially replayed. Each star represents a synapse in the direction of S1 (left) or S1* (right). Dashed line indicates the threshold (66% of Up states) used to identify synapses that are replayed reliably for analysis in (B); purple line indicates the maximum number of Up states; blue line demarcates the 50% mark of the total number of Up states. (**B**) Thresholded connectivity matrix indicating synaptic connections showing reliable replays for S1 (blue) or S1* (red). Grey boxes highlight between group connections. (**C**) Number of replay events for inter-group synapses per Up state across all Up states (left) and a subset of Up states (right) for S1 (blue) and S1* (red). Note that

*Figure 6 continued on next page*

*Figure 6 continued*

both sequences show similar high number of replays across all Up states, suggesting that both sequences are replayed during each Up state. (D) Network's graphs showing between group (top/middle) and within group (bottom) connections after sleep. Edges shown here are those which revealed reliable replays of S1 (blue) and S1* (red) as shown in B (right). Nodes are colored blue (red) if more than 50% of their incoming connections show reliable replay in direction of S1 (S1*). Green nodes indicate neurons with high in-degrees, receiving the same number of 'replayed' synapses from left and right, and black indicates that none of these conditions are met.

S1*) and those participating in both sequences replays (see *Figure 6B*, where some target neurons (X-axis) receive input from source neurons (Y-axis) in only one network 'direction', left (blue) or right (red), and others receive input from both 'directions').

To confirm that both memories are replayed within the same Up state (i.e., some synapses replay S1 and others replay S1* during a given Up state), we counted, for each Up state, the total number of individual replay events across all synapses that were identified to replay S1 and S1* reliably (*Figure 6C*). This revealed fluctuations from one Up state to another, but the count remained high for both S1 and S1* confirming our prediction that partial replays of both sequences occur during the same Up state, that is, any given Up state participates in replay of both memories. Still, zoom-in to the replay count diagram (*Figure 6C*, right) revealed an antiphase oscillation, that is, one Up state would replay more S1 synapses, while another one (commonly next one) would replay more S1* synapses. Note, our model predicts that partial sequences (specifically spike doubles) of both memories can be replayed during *the same Up state* and not that both are replayed *simultaneously* (at the same exact time). Comparing replays during first vs second half of an Up state, we found that more replay events happened during the first half of any given Up state (particularly near the Down to Up transition) compared to the second half (not shown). This result is consistent with electrophysiological data suggesting that memory replay is strongest at the Down to Up state transition (*Johnson et al., 2010*).

Finally, in *Figure 6D*, we plotted all the synapses identified by the analysis in *Figure 6B*, that is, those involved in reliable (in more than 66% of all Up states) replay during sleep: top plot shows synapses in S1 direction (in blue) and bottom one shows synapses in S1* direction (in red). For each neuron we compared the number of such synapses it received from its left (S1 direction) vs right (S1* direction) neighboring population (e.g., for a neuron in group B, we compared if it received more synapses demonstrating reliable replay from group A or from group C). We then colored in blue (red) neurons receiving more synapses demonstrating reliable replay from its left (right) neighbors (*Figure 6D*). In green we colored neurons receiving the same number of 'replayed' synapses from left and right. While we found that many neurons (blue or red) participated reliably in only one sequence replay, S1 or S1*, a few neurons (green) participated equally in replay of both sequences, creating 'network hubs'.

## Sleep replay leads to competition between synapses

In order to further understand how sleep replay affects S1 and S1* memory traces to allow enhancement of both memories, we next analyzed the dynamics of individual synaptic weights within the population of neurons containing the overlapping memory sequences (i.e. neurons 200–249). *Figure 7A* shows distributions of synaptic weights for synapses in the direction of S1 (top row) and in the direction of S1* (bottom row) before (blue) and after (red) specific events. Different columns correspond to different events, i.e. after S1 training (*Figure 7A*, left), after S1* training (*Figure 7A*, middle), after sleep (*Figure 7C*, right). Prior to any training, synaptic weights in the direction of either memory sequence were Gaussian distributed (*Figure 7A*, blue histogram, left). After S1 training, the weights for S1 strengthened (shifted to the right), while the weights for S1* weakened (shifted to the left). As expected, this trend was reversed when S1* was trained (*Figure 7A*, middle). After sleep, for each sequence (S1 or S1*) there was a subset of synapses that were further strengthened, while the rest of synapses were weakened (*Figure 7A*, right). This suggests that sleep promotes competition between synapses, so that specific subsets of synapses uniquely representing each memory trace can reach the upper bound to maximize recall performance while other synapses would become extinct to minimize interference.

Because of the random 'anatomical' connectivity, the cortical network model included two classes of synapses: *recurrent/bidirectional*, when a pair of neurons (e.g., n1 and n2) are connected by

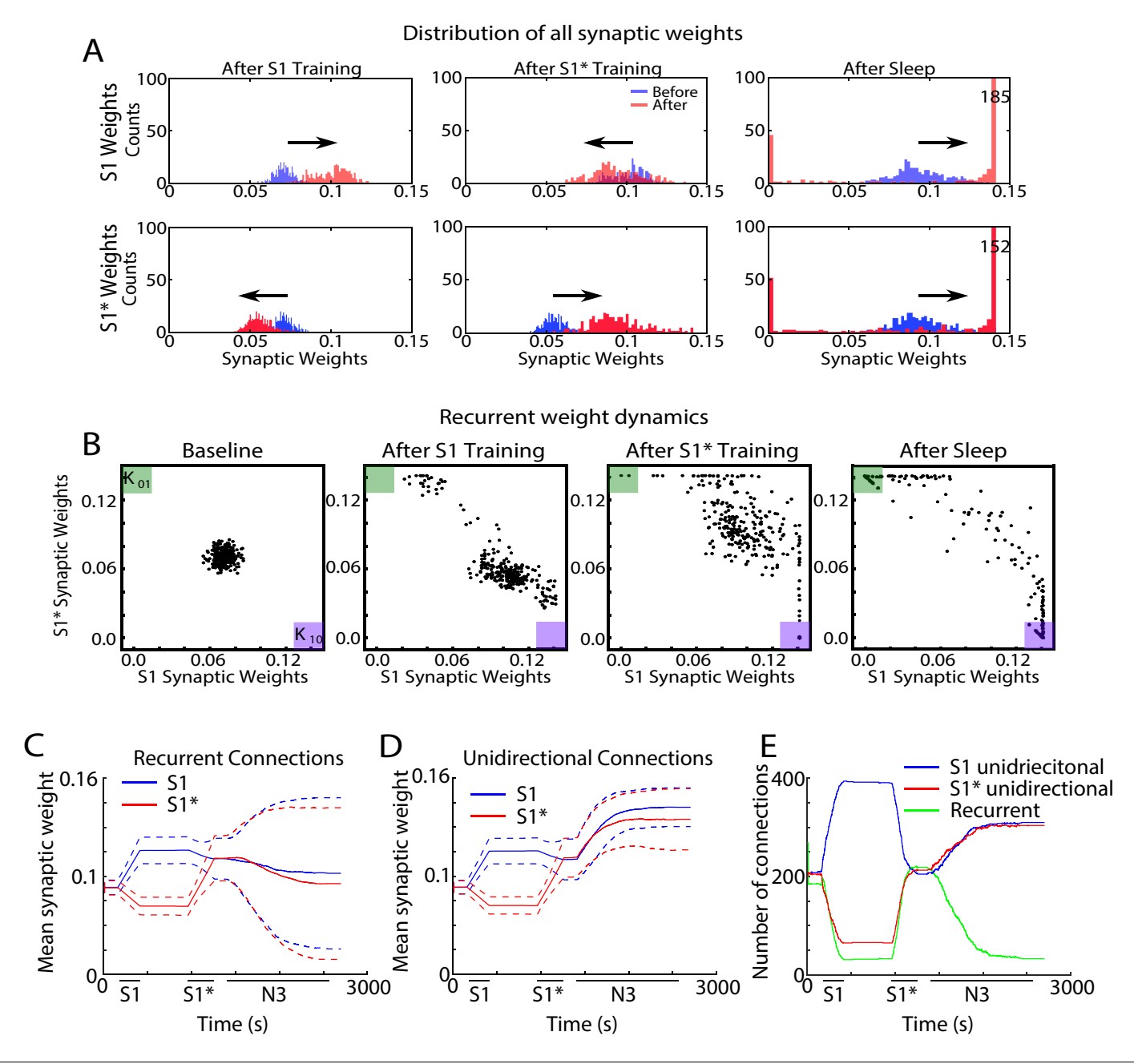

**Figure 7.** Sleep promotes unidirectional synaptic connectivity with different subsets of synapses becoming specific to the old or new memory sequences. (**A**) Dynamics of synaptic weight distributions from the trained region. Top row shows strength of synapses in direction of S1. Bottom row shows strength of synapses in direction of S1*. Blue shows the starting points for weights, and red shows new weights after different specific events, for example, S1 training, S1* training, sleep. (**B**) Scatter plots show synaptic weights for all reciprocally connected pairs of neurons before and after training (left/middle) and after sleep (right). For each pair of neurons (e.g., n1-n2), the X-coordinate shows the strength of $W_{n1 \to n2}$ synapse and the Y-coordinate shows the strength of $W_{n2 \to n1}$ synapse. The green ($K_{01}$) and purple ($K_{10}$) boxes show the locations in the scatter plot representing synaptic pairs with strong preference for S1* (green) or S1 (purple). (**C**) The evolution of the mean synaptic strength (solid line) and the standard deviation (dashed line) of recurrent connections in S1 (blue) and S1* (red) direction. Note the large standard deviation after sleep indicating strong synaptic weight separation, so each recurrent neuronal pair supports preferentially either S1 or S1*. (**D**) The evolution of the mean synaptic weight (solid line) and the standard deviation (dashed line) of unidirectional connections in S1 (blue) and S1* (red) direction. Note the overall increase in synaptic strength after sleep. (**E**) The number of functionally recurrent and unidirectional connections in the trained region of the network as a function of time, obtained after thresholding the network connectivity matrix with threshold 0.065 (which is smaller than the initial mean synaptic strength). Note the decrease of functionally recurrent connections and increase of functionally unidirectional connections after sleep.

*Figure 7 continued on next page*

*Figure 7 continued*

The online version of this article includes the following figure supplement(s) for figure 7:

**Figure supplement 1.** Interleaved training revealed synaptic weight dynamics that are similar to sleep but result in less segregation of synaptic weights.

**Figure supplement 2.** Synaptic plasticity that is biased towards LTP or LTD also results in memory orthogonalization during sleep .

opposite synapses (n1→ n2 *and* n2→ n1) and *unidirectional* (n1→ n2 *or* n2→ n1). In the following we looked separately at these two classes. We also compared synaptic weights dynamics during sleep (*Figure 7*) vs interleaved training (*Figure 7—figure supplement 1*).

In the scatter plots of synaptic weights for the recurrent synapses (*Figure 7B*), for each pair of neurons (e.g., n1-n2), we plotted a point with the X-coordinate representing the weight of n1→ n2 synapse and the Y-coordinate representing the weight of n2→ n1 synapse. Any point with X- (Y-) coordinate close to zero would, therefore, indicate a pair of neurons with functionally unidirectional coupling in S1* (S1) direction. The initial Gaussian distribution of weights (*Figure 7B*, left) was pushed towards the bottom right corner of the plot ($K_{10}$, purple box), indicating increases in S1 weights and relative decrease of S1* weights in response to S1 training (*Figure 7B*, middle/left). It should be noted that a small subset of synaptic weights increased in the direction of S1* during S1 training. Analysis of this population of synaptic weights revealed that these connections were comprised solely of 'within group' connections. It is important to note that these synapses did not impair the consolidation of the trained memory but instead helped to increase activity *within each group* regardless of which sequence was recalled.

Training of S1* caused an upward/left shift representing strengthening of S1* weights and weakening of S1 weights (*Figure 7B*, middle/right). For very long S1* training (not shown) almost all the weights would be pushed to the upper left corner ($K_{01}$, green box). Sleep appears to have taken most of the weights located in the center of the plot (i.e., strongly bidirectional synapses) and separated them by pushing them to the opposite corners ($K_{01}$, green box, and $K_{10}$, purple box) (*Figure 7B*, right). In doing so, sleep effectively converted recurrent connections into unidirectional connections which preferentially contributed to one memory sequence or another. It should be noted that interleaved training resulted in similar separation of weights such that some previously recurrent synapses became functionally unidirectional (*Figure 7—figure supplement 1A, B*). Interleaved training, however, retained more recurrent weights than sleep likely contributing to the smaller improvement in performance during post-interleaved training testing (*Figure 4—figure supplement 1C*).

Sleep-dependent synaptic weight dynamics are further illustrated in *Figure 7* panels C-E. The mean strength of all recurrent connections in the trained region decreased slightly during sleep (*Figure 7C*), however the standard deviation increased significantly (see dashed lines in *Figure 7C*). The last reflected strong asymmetry of the connection strength for recurrent pairs after sleep, again indicating that sleep effectively converts recurrent connections into unidirectional ones. Indeed, the mean strength of all unidirectional connections increased during sleep (*Figure 7D*, blue and red lines). We next counted the total number of functionally recurrent and unidirectional connections after training and after sleep (*Figure 7E*). In this analysis if one branch of a recurrent pair reduced in strength below the threshold, it was counted as unidirectional. After sleep, the number of recurrent connections dropped to just about 15% of what it was after training. Interleaved training resulted in similar but smaller changes to unidirectional and bidirectional connections (*Figure 7—figure supplement 1C, D, E*). Together these results suggest that sleep decreases the density of recurrent connections and increases the density of unidirectional connections, both by increasing the *strength* of anatomical unidirectional connections and by *converting* anatomical recurrent connections to functionally unidirectional connections. This allows the assignment of individual neurons to unique memories, that is, orthogonalization of memory representations, so that multiple memories could replay without interference during the same Up states of slow oscillations and can be recalled successfully after sleep.

## LTP or LTD biased synaptic plasticity still leads to orthogonalization of memory representations during sleep

In all previous simulations, LTP and LTD were balanced during sleep and interleaved training. To test that the orthogonalization of the memory traces during sleep is independent of the specific balance of LTP/LTD ($A_+/A_-$, see *Methods and Materials*), we performed additional simulations biasing the LTP/LTD ratio during sleep towards either LTD (*Figure 7—figure supplement 2A*; $A_+/A_-$=0.0019/0.002) or LTP (*Figure 7—figure supplement 2B*; $A_+/A_-$=0.0021/0.002). We found that in both cases, sleep resulted in the orthogonalization of memory representations. Scatter plots of bidirectional synaptic connections (the same analysis as in *Figure 7B*) revealed that sleep formed strongly memory specific configurations of weights by pushing some of the recurrent connections to either the top left (S1* preferential) or bottom right (S1 preferential) corners of the plot. The red lines on these plots depict the threshold used to identify neuronal pairs that are either strongly preferential for S1 (bottom right corner) or S1* (top left corner). The number of synapses above these thresholds were quantified in the bar plots below showing that sleep increases the density of the memory specific connections between neurons regardless of the LTP/LTD ratio (*Figure 7—figure supplement 2*, bottom panels). The vector field plots (*Figure 7—figure supplement 2*, middle panels) provide a summary of the average synaptic weight dynamics during training (left and middle plots) and during sleep (right plot). It revealed convergence towards the corners (note arrows pointing to the corners during sleep phase) which represent cell pairs being strongly enrolled either to sequence S1 or sequence S1* encoding.

It should be noted that because our model does not have homeostatic mechanisms to regulate 'average' synaptic strength during sleep, the case of LTD biased sleep revealed a net reduction of synaptic strengths, while the LTP biased condition showed a net increase. For LTD biased sleep, many recurrent synapses decreased the strength while a fraction of synapses kept or even increased the strength. These synapses became memory specific after sleep. This observation may be in line with ideas from Tononi and colleagues showing net reductions of synaptic weights during sleep (*Tononi and Cirelli, 2014*) however, more analysis of the model including additional homeostatic rules is need to make this conclusion based on model simulations.

## Neurons participating in sleep replay are the same as those responding earlier during memory recall

In the previous sections, we found that for overlapping memories sleep leads to segregation of the entire population of neurons into two subsets based on (a) asymmetric synaptic input from left/right neighboring groups (e.g., subset $B_i$ of neurons from group B receives stronger total synaptic input from group A compare with total input from group C; subset $B_j$ of neurons from group B receives stronger input from C than from A) (*Figure 5D,E*); (b) preference to participate reliably in only one specific sequence replay during sleep (e.g., subset $B_k$ of neurons from group B receives more synapses demonstrating reliable replay from group A than from group C; this is reversed for subset $B_l$ of neurons from group B) (*Figure 6D*). Here we tested if these groups of neurons, identified by synaptic strength and replay, overlap. We also compared them to the subset of neurons responding earliest within each group during memory recall.

Instead of stimulating only groups A or E, here we stimulated independently every single group - A, B, C, D, E (*Figure 8A*). We then obtained the response delay for each neuron in groups B, C, D when its respective left vs right neighboring groups were stimulated, and we calculated the difference of delays. Thus, for example, we measured a difference of response delays for each neuron in group B when either group A or group C was stimulated. This analysis is similar to what was done in (*Navratilova et al., 2012*), where the difference of place cell responses at a specific location on a linear track was calculated when a rat was approaching that location from one direction vs the other. *Figure 8B* shows the distribution of delays at different times. As expected, it became asymmetric after S1 training (e.g., in group B more neurons responded earlier upon stimulation of group A vs stimulation of group C), symmetric again after S1* training, and finally symmetric but wider after sleep. The last suggests that sleep increases segregation of neurons into two groups specific to each memory based on response delay (e.g., in group B some neurons, $B_n$, responded earlier upon stimulation of group A vs stimulation of group C; while other neurons, $B_m$, responded earlier upon stimulation of group C vs stimulation of group A). Indeed, the directionality index based on delays

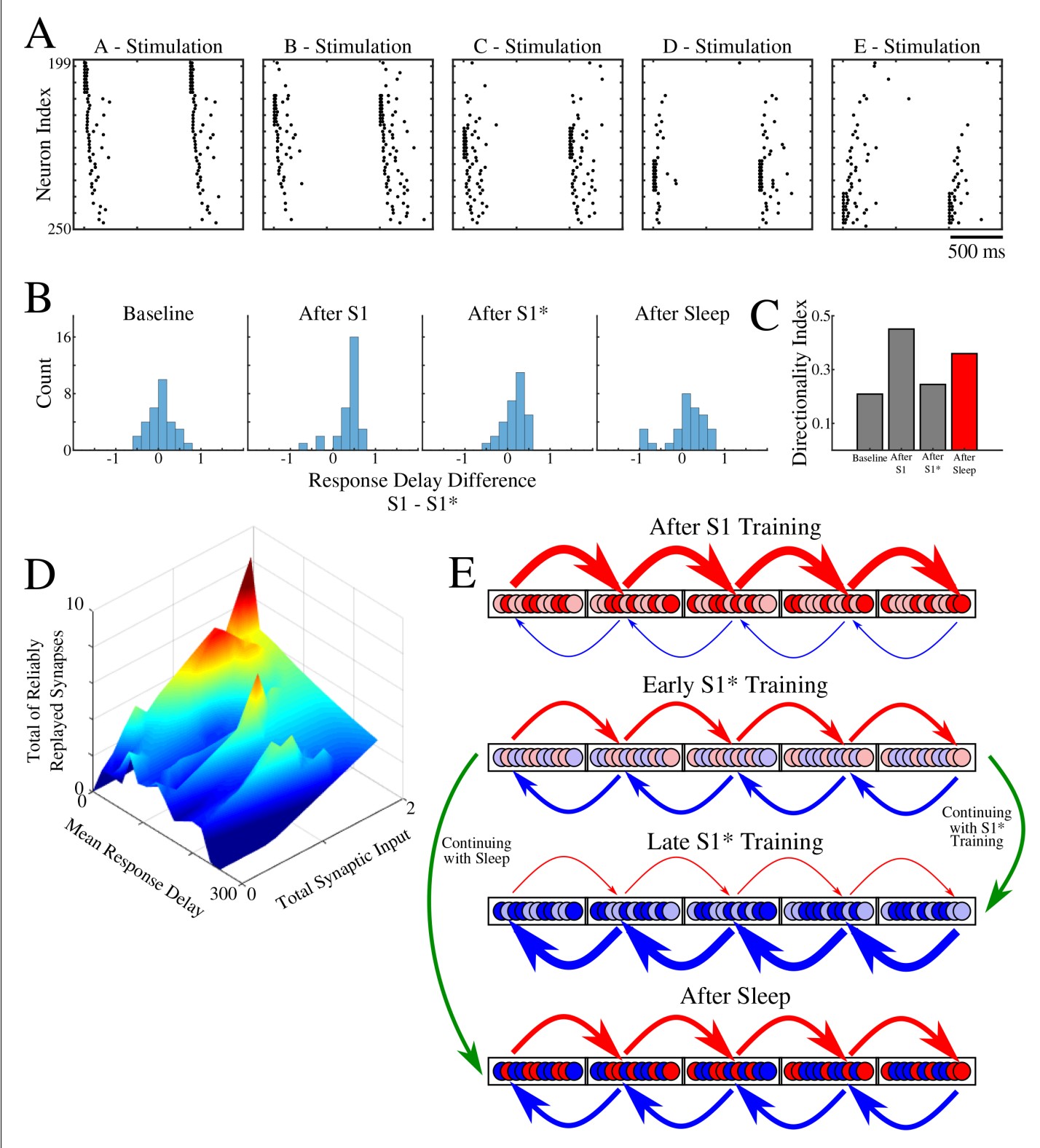

**Figure 8.** Population of neurons participating in reliable replay during sleep overlaps with the early responders during memory recall. (**A**) Characteristic examples of the network activity showing spiking events during stimulations of each individual 'letter' of a memory sequence in awake. (**B**) Distributions of the differences in response delay for all neurons from the trained region when the respective left vs right neighboring groups are stimulated, as shown in **A**. (**C**) Response delay-based directionality index before/after training of each sequence (S1/S1*) in gray, and after sleep (red). (**D**) 3-D surface plot showing, for each neuron from a trained region, the number of incoming synapses demonstrating reliable replays during sleep (z-axis), mean
*Figure 8 continued on next page*

*Figure 8 continued*

response delay during testing (as in panel A) after sleep (y-axis), and total synaptic input to a neuron after sleep (x-axis). Note that neurons receiving highest total synaptic inputs in specific network direction after sleep are also those who respond with shortest delay during testing recall in that direction after sleep and also those who receive the highest number of synapses demonstrating reliable replay during sleep. (E) Simplified cartoon of the network connectivity after different training phases followed by sleep. Arrows indicate connections between neurons (nodes) with blue arrows being connections strong for S1 and red for S1*. Blue and red nodes represent neurons that contribute (weakly - light colors; strongly - dark colors) to recall of S1 and S1*, respectively. Top, network configuration after S1 training – all nodes and connections are allocated to S1. Middle/Top, network configuration after initial S1* training – nodes/connections start to learn S1* and 'unlearn' S1. The information about the old memory S1 is still available. Middle/Bottom, network configuration after continuing S1* training – all nodes/connections are allocated to S1*. All information about S1 is lost. Bottom, network configuration when initial S1* training is followed by sleep – orthogonalization of memory traces, some nodes/connection are allocated to S1 and others to S1*.

(*Figure 8C*) revealed an increase after S1 training, drop after S1* training, and increase again after sleep.

In *Figure 8D*, we summarized our results by putting together three main characteristics we discussed in this study: Mean response delay of a neuron during stimulation of its neighboring group, Total synaptic input a neuron receives from that neighboring group, and Number of connections to a neuron from that neighboring group that are replayed reliably during sleep. We found a strong correlation between these three measures, that is, the neurons who responded with a shortest delay during a given sequence recall after sleep are the same neurons who received strongest synaptic input in that sequence direction after sleep and were involved in most of that sequence replays during sleep.

Together, our study proposes the following network connectivity dynamics during learning and sleep (*Figure 8E*). Initial training allocates all available neuronal/synaptic resources to a single memory (S1) (*Figure 8E*, top); some neurons contribute stronger than others (light vs dart colors in *Figure 8E*; based on *Figure 7B*). Subsequent training of a competing memory (S1*) progressively erases the initial memory trace by reallocating synaptic resources to the new memory; an initial segregation of neurons is formed (*Figure 8E*, middle/top). Continuing training of a competing memory (S1*) leads to complete and irreversible damage to the old memory (S1) – catastrophic forgetting (*Figure 8E*, middle/bottom). A sleep phase implemented before the old memory is erased allows replay of both old and new memory traces; this divides resources between competing memories leading to the formation of the orthogonal memory representations which allows the co-existence of multiple memories within overlapping populations of neurons (*Figure 8E*, bottom).

## Discussion

We report here that sleep can reverse catastrophic forgetting of previously learned (old) memories after damage by new training. Sleep is able to accomplish this task through spontaneous reactivation (replay) of both old and new memory traces, leading to reorganization and fine-tuning of synaptic connectivity. As a result, sleep creates unique orthogonal representations of the competing memories that allow their co-existence without interference within overlapping ensembles of neurons. Thus, if without competition, a memory is represented by the entire available population of neurons and synapses, in the presence of competition, its representation is reduced to a *subset* of neurons/synapses which selectively encode a given memory trace. Our study predicts that memory representations in the brain are dynamic; after each new episode of training followed by sleep, the synaptic representations of the old memories, sharing resources with the new task, may change to achieve an optimal separation among the memory traces occupying overlapping ensembles of neurons. Our study suggests that sleep, by being able to directly reactivate memory traces encoded in synaptic weight patterns, provides a powerful mechanism to prevent catastrophic forgetting and enable continual learning.

### Catastrophic forgetting and continual learning

The work on catastrophic forgetting and interference in connectionist networks was pioneered by *Mccloskey and Cohen, 1989* and *Ratcliff, 1990*. Catastrophic interference is observed when a previously trained network is required to learn new data, e.g., a new set of patterns. When learning new

data, the network can suddenly erase the memory of the old, previously learned inputs (*French, 1999*; *Hasselmo, 2017*; *Kirkpatrick et al., 2017*). Catastrophic interference is thought to be related to the so-called 'plasticity-stability' problem. This problem comes from the difficulty of creating a network with connections which are plastic enough to learn new data, while stable enough to prevent damage to the old memories. Due to the inherent trade-off between plasticity and memory stability, catastrophic interference and forgetting remains to be a difficult problem to overcome in connectionist networks (*French, 1999*; *Hasselmo, 2017*; *Kirkpatrick et al., 2017*).

A number of attempts have been made to overcome catastrophic interference (*French, 1999*; *Hasselmo, 2017*; *Kirkpatrick et al., 2017*). Early attempts included changes to the backpropagation algorithm, implementations of a 'sharpening algorithm' in which a decrease in the overlap of internal representations was achieved by making hidden-layer representations sparse, or changes to the internal structure of the network (*French, 1999*; *Hasselmo, 2017*; *Kirkpatrick et al., 2017*). These attempts were able to reduce the severity of catastrophic interference in specific cases but could not provide a complete and generic solution to the problem. Another popular method for preventing interference and forgetting is to explicitly retrain or rehearse all the previously learned inputs while training the network on the new data – interleaved training (*Hasselmo, 2017*). This idea recently led to a number of successful algorithms to constrain the catastrophic forgetting problem, including interleaved training focusing on the previously known items overlapping with new training data (*McClelland et al., 2020*), generative algorithms to generate previously experienced stimuli during the next training period (*Zz and Hoiem, 2018*; *van de Ven and Tolias, 2018*) and generative models of the hippocampus and cortex to generate examples from a distribution of previously learned tasks in order to retrain (replay) these tasks during a sleep phase (*Kemker and Kanan, 2017*).

In agreement with these previous studies, we show that interleaved training can prevent catastrophic forgetting resulted from sequential training of the overlapping spike patterns. This method, however, does not necessarily achieve optimal separation between old and new overlapping memory traces. Indeed, interleaved training requires repetitive activation of the entire memory patterns, so if different memory patterns compete for synaptic resources (as for the opposite sequences studied here) each phase of interleaved training will enhance one memory trace but damage the others. This is in contrast to replay during sleep when only memory specific subsets of neurons and synapses may be involved in each replay episode. Another primary concern with interleaved training is that it becomes increasingly difficult/cumbersome to retrain all the memories as the number of stored memories continues to increase and access to the earlier training data may no longer be available. As previously mentioned, biological systems have evolved a mechanism to prevent this form of forgetting without the need to explicitly retrain the network on all previously encoded memories. Studying how biological systems overcome catastrophic forgetting can provide insights into novel techniques to combat this problem in artificial neural networks.

## Sleep and memory consolidation

Though a variety of sleep functions remain to be understood, there is growing evidence for the role of sleep in consolidation of newly encoded memories (*Paller and Voss, 2004*; *Walker and Stickgold, 2004*; *Oudiette et al., 2013*; *Rasch and Born, 2013*; *Stickgold, 2013*; *Weigenand et al., 2016*; *Wei et al., 2018*). The mechanism by which memory consolidation is influenced by sleep is still debated, however a number of hypotheses have been put forward. One such hypothesis is the 'Active System Consolidation Hypothesis' (*Rasch and Born, 2013*). Central to this hypothesis is the idea of repeated memory reactivation (*Wilson and McNaughton, 1994*; *Skaggs and McNaughton, 1996*; *Paller and Voss, 2004*; *Mednick et al., 2013*; *Oudiette et al., 2013*; *Oudiette and Paller, 2013*; *Rasch and Born, 2013*; *Stickgold, 2013*; *Weigenand et al., 2016*). Although NREM sleep was shown to be particularly important for reactivation of declarative (hippocampus-dependent) memories (*Marshall et al., 2006*; *Mednick et al., 2013*), human studies suggest that NREM sleep may be also involved in the consolidation of the procedural (hippocampus-independent) memories. This includes, for example simple motor tasks (*Fogel and Smith, 2006*), or finger-sequence tapping tasks (*Walker et al., 2002*; *Laventure et al., 2016*). Selective deprivation of NREM sleep, but not REM sleep, reduced memory improvement for the rotor pursuit task (*Smith and MacNeill, 1994*). Following a period of motor task learning, the duration of NREM sleep (*Fogel and Smith, 2006*) and the number of sleep spindles (*Morin et al., 2008*) increased. The amount of performance increase in the finger tapping task correlated with the amount of NREM sleep (*Walker et al., 2002*), spindle

density (*Nishida and Walker, 2007*) and delta power (*Tamaki et al., 2013*). In a recent animal study (*Ramanathan et al., 2015*), consolidation of the procedural (skilled upper-limb) memory depended on bursts of spindle activity and slow oscillations during NREM sleep.

## Model predictions

The model of awake training and sleep consolidation presented in our new study was designed to simulate learning and consolidation of procedural memory tasks. Indeed, in our model, training a new task directly impacts cortical synaptic connectivity that may be already allocated to other (old) memories. We found that as long as damage to the old memory is not sufficient to completely erase its synaptic footprint, sleep can enable replay of both old and newer memory traces and reverse the damage while improving performance. Thus, to avoid irreversible damage, new learning in our model is assumed to be slow which may correspond to learning a procedural task, for example, new motor skill, over multiple days allowing sleep to recover old memory traces that are damaged by each new episodes of learning.

Nevertheless, we suggest that our model predictions, at least at the synaptic level, are not limited to a specific type of memory (declarative vs procedural) or specific type of sleep (NREM vs REM). Replay during REM sleep (*Louie and Wilson, 2001*) may trigger synaptic weight dynamics similar to that we described here. Though REM is characterized by less synchronized spiking activity, the occurrence of memory replay during REM is supported by place cell recordings (*Louie and Wilson, 2001*) and electroencephalography studies in humans (*Atienza and Cantero, 2001*). While synchronized activity is helpful for replay and may allow (because of high spike precision) for replay to occur at compressed time scales, as observed during NREM sleep (*Euston et al., 2007*), the crucial component of replay is the defined spike ordering which may be happening during REM sleep even when the overall network synchronization is low. Indeed, we observed similar synaptic weight dynamics and orthogonalization of memory representations when periodic Up/Down state oscillations were replaced by continuous REM-like spiking activity. While our model lacks hippocampal input, we showed previously (*Wei et al., 2016*; *Sanda et al., 2019*) that sharp wave-ripple (SWR) like input to the cortical network would trigger replay of previously learned cortical sequences during SWS. This suggests, in agreement with (*Skelin et al., 2019*), that replay driven by hippocampal inputs may reorganize the cortical synaptic connectivity in a matter similar to spontaneous replay we described here.

Our model predicts the possibility of the partial sequence replays, that is, when short snippets of a sequence are replayed independently, within the cortex. Furthermore, we showed that reliable partial replays of overlapping memory traces can occur during the same cortical Up state. That is to say, during an Up state rather than replaying the entire sequence A-B-C-D-E, we observed replay of individual transitions (e.g. A-B, D-E, C-D). We can speculate that for strongly overlapping sequences, as we modeled in this study, such partial replay would allow to replay snippets of both sequences with less interference during the same Up state. Indeed, recent data (*Ghandour et al., 2019*) have shown evidence for partial memory replay during NREM sleep (also see *Swanson et al., 2020*).

Importantly, our model of sleep consolidation predicts that the critical synaptic weight information from previous episodes of learning is still preserved after new training even if recall performance for the older task is significantly reduced. Because of this, spontaneous activity during sleep combined with unsupervised plasticity can trigger reactivation of the previously learned memory patterns and modify synaptic weights reversing damage from the new learning. It further suggests that the apparent loss of performance for older tasks in the artificial neuronal networks after new training – catastrophic forgetting – may not imply irreversible loss of information as it is generally assumed. Indeed, our recent work (*Krishnan et al., 2019*) revealed that simulating a sleep-like phase in feedforward artificial networks trained using backpropagation can provide a solution for the catastrophic forgetting problem in agreement with our results from the biophysical model presented here. Few changes to the network properties, normally associated with transition to sleep, were critical to accomplish this goal: relative hyperpolarization of the pyramidal neurons and increasing strength of excitatory synaptic connectivity. Both are associated with known effects of neuromodulators during wake-sleep transitions (*McCormick, 1992*) and were previously implemented in the thalamocortical model (*Krishnan et al., 2016*) that we used in our new study. Interestingly, these changes would make neurons relatively less excitable and, at the same time, increase contribution of the strongest synapses, effectively enhancing the dynamical range for the trained synaptic patterns and reducing

contribution of synaptic noise; together this would promote replay of the previously learned memories.

The 'Sleep Homeostasis Hypothesis' (*Tononi and Cirelli, 2014*) suggests that homeostatic mechanisms active during sleep should result in a net synaptic depression to renormalize synaptic weights and to stabilize network dynamics. In our model, LTP and LTD were generally balanced during sleep and no homeostatic mechanisms were implemented to control net synaptic dynamics. However, when synaptic plasticity during sleep was explicitly biased towards LTD, sleep was still able to selectively increase a subset of synaptic weights, thus making them memory specific, while reducing the strength of other synapses. We predict that this mechanism may aid in increasing the memory capacity of the network by only strengthening the minimal number of connections required for the preservation of memories and resetting other synapses towards baseline strength during sleep. The network would then be able to use these synapses to encode new memories thus potentially facilitating continual learning without the consequence of retroactive interference.

## Comparison to experimental data and model limitations

There are evidences that memory replay during SWS occurs predominantly near the Down to Up state transitions (*Johnson et al., 2010*). This observation comes from *in vivo* studies in which multiple brain regions, including the hippocampus and cortex, are in continual communication during SWS. It has been shown that sharp-wave ripples tend to occur at the Down to Up state transition (*Sanda et al., 2019*; *Skelin et al., 2019*), which may explain the predomiance of the hippocampus driven replay at the beginning of cortical Up states. We did not explicitly model the hippocampus or hippocampal inputs in our study. Rather, we assumed that memory traces are already embedded to the cortical connectivity matrix either because of the earlier hippocampal-dependent consolidation or because these memories are hippocampal-independent (as for procedural memories). We found that such cortical memory traces also tend to replay more during the initial phase of an Up state, possibly because of the higher firing rate, but replay continues throughout the entire Up state duration. This predicts that hippocampal dependent replay of new memories, that are not yet encoded in the cortex, may occur earlier in the Up state compared to the spontaneous replay of the old memory traces, which may occur later in the Up state.

Our results are consistent with *in vivo* experiments with rats running on a linear track and we make several specific predictions for future experiments. Specifically, our model predicts: (a) running in one direction on a linear track would lead to backwards receptive field expansion (confirmed for hippocampus [*Mehta et al., 1997*]); (b) forwards and backwards running on a linear track would lead to developing asymmetric receptive fields for different neurons (confirmed for hippocampus [*Navratilova et al., 2012*]); (c) running on a belt track in a VR apparatus first only in one direction and then in reverse one could damage the learning associated with first task; (d) SWS implemented after training would reverse damage and further enhance task specificity of neurons.

It is important to note that (*Mehta et al., 1997*) found that backwards expansion of the place fields was reset between sessions. Later, (*Roth et al., 2012*) found that the resetting of the backwards expanded place fields between sessions was a phenomenon specific to the CA1 and place fields did not reset in CA3. These results suggest that synapses in CA3 vs CA1 may have different plasticity properties. Furthermore, the neocortex may have entirely different synaptic dynamics since its goal is long term storage as opposed to temporal memory encoding. With successive sleep periods, cortical memories become hippocampal independent (*Lehmann and McNamara, 2011*) and this may explain why resetting of the place fields was observed in CA1 (*Mehta et al., 1997*). Our study predicts that the cortical (such as associate cortex) representations of the sequence memories undergo a similar form of backwards expansion as it was observed in CA1. This form of backwards expansion, however, persists and even increases after sleep.

The phenomena of backwards memory replay and decrease in number of memory replays over time have been observed in rat hippocampus for recent memories. Within the hippocampus, backwards replay is predominantly observed during a post-task awake resting period (*Foster and Wilson, 2006*). The studies of hippocampal replay (*O'Neill et al., 2008*; *Giri et al., 2019*) found decreases in replay of familiar sequences over time, which may occur because of the hippocampal SWRs inducing persistent synaptic depression within the hippocampus (*Norimoto et al., 2018*). We did not observe backwards replay; rather, forward replay in the model persisted during sleep. However, we believe there is no definitive evidence for either backwards replay or decrease in memory

replays in the cortex. The opposite, in fact, may be true. Cortical replay of recently formed memories results in a tagging of synapses involved in consolidation of those memories by increasing their synaptic efficacy (*Langille, 2019*). These tagged synapses may likely be reactivated throughout sleep thereby resulting in more cortical replay during both NREM and REM sleep (*Diekelmann and Born, 2010*; *Langille, 2019*).

To summarize, our study predicts that sleep could prevent catastrophic forgetting and reverse memory damage through replay of old and new memory traces. By selectively replaying new and competing old memories, sleep dynamics not only achieve consolidation of new memories but also provide a mechanism for reorganizing the synaptic connectivity responsible for previously learned memories – re-consolidation of old memory traces – to maximize separation between memory representations. By assigning different subsets of neurons and synapses to primarily represent different memory traces, sleep effectively orthogonalizes memory representations to allow for overlapping populations of neurons to store competing memories and to enable continual learning.

# Materials and methods

## Thalamocortical network model

### Network architecture

The thalamocortical network model used in this study has been previously described in detail (*Krishnan et al., 2016*; *Wei et al., 2016*; *Wei et al., 2018*) and the code is available in (https://github.com/o2gonzalez/sequenceLearningSleepCode; copy archived at https://github.com/elifesciences-publications/sequenceLearningSleepCode; *González, 2020b*). Briefly, the network was comprised of a thalamus which contained 100 thalamocortical relay neurons (TC) and 100 reticular neurons (RE), and a cortex containing 500 pyramidal neurons (PY) and 100 inhibitory interneurons (IN). The model contained only local network connectivity as described in *Figure 1*. Excitatory synaptic connections were mediated by AMPA and NMDA connections, while inhibitory synapses were mediated through $GABA_A$ and $GABA_B$. Starting with the thalamus, TC neurons formed AMPA connections onto RE neurons with a connection radius of 8 ($R_{AMPA(TC-RE)}$=8). RE neurons then projected inhibitory $GABA_A$ and $GABA_B$ connections onto TC neurons with $R_{GABA-A(RE-TC)}$=8 and $R_{GABA-B(RE-TC)}$=8. Inhibitory connections between RE-RE neurons were mediated by $GABA_A$ connections with $R_{GABA-A(RE-RE)}$=5. Within the cortex, PY neurons formed AMPA and NMDA connections onto PYs and INs with $R_{AMPA(PY-PY)}$=20, $R_{NMDA(PY-PY)}$=5, $R_{AMPA(PY-IN)}$=1, and $R_{NMDA(PY-IN)}$=1. PY-PY AMPA connections had a 60% connection probability, while all other connections were deterministic. Cortical inhibitory IN-PY connections were mediated by $GABA_A$ with $R_{GABA-A(IN-PY)}$=5. Finally, connections between thalamus and cortex were mediated by AMPA connections with $R_{AMPA(TC-PY)}$=15, $R_{AMPA(TC-IN)}$=3, $R_{AMPA(PY-TC)}$=10, and $R_{AMPA(PY-RE)}$=8.

### Wake - Sleep transition

To model the transitions between wake and sleep states the model included synaptic and intrinsic mechanisms which reflect the changes in neuromodulatory tone during these different arousal states as previously described in *Krishnan et al., 2016*. We included the effects of acetylcholine (ACh), histamine (HA), and GABA. ACh modulated potassium leak currents in all neuron types and excitatory AMPA connections within the cortex only. HA modulated the activation of the hyperpolarization-activated mixed cation current in TC neurons only, and GABA modulated the strength of inhibitory GABAergic synapses in both cortex and thalamus. As compared to the awake state, the levels of ACh and HA were reduced during NREM slow wave sleep, while the level of GABA was increased. This was done to reflect experimental observations of changes in the relative concentrations of ACh, HA, and GABA during different sleep stages (*Vanini et al., 2012*).

### Intrinsic currents

All neurons were modeled with Hodgkin-Huxley kinetics. Cortical PY and IN neurons contained dendritic and axo-somatic compartments as previously described (*Wei et al., 2018*). The membrane potential dynamics were modeled by the following equation:

$$C_m \frac{dV_D}{dt} = -I_D^{Na} - I_D^{NaP} - I_D^{Km} - I_D^{KCa} - ACh_{gkl}I_D^{KL} - I_D^{HVA} - I_D^L - g(V_D - V_S) - I^{syn},$$

$$g(V_D - V_S) = -I_S^{Na} - I_S^{NaP} - I_S^K,$$

where $C_m$ is the membrane capacitance, $V_{D,S}$ are the dendritic and axo-somatic membrane voltages respectively, $I^{Na}$ is the fast sodium (Na$^+$) current, $I^{NaP}$ is the persistent Na$^+$ current, $I^{Km}$ is the slow voltage-dependent non-inactivating potassium (K$^+$) current, $I^{KCa}$ is the slow calcium (Ca$^{2+}$)-dependent K$^+$ current, $ACh_{gkl}$ represents the change in K$^+$ leak current $I^{KL}$ which is dependent on the level of acetylcholine (ACh) during the different stages of wake and sleep, $I^{HVA}$ is the high-threshold Ca$^{2+}$ current, $I^L$ is the chloride (Cl$^-$) leak current, $g$ is the conductance between the dendritic and axo-somatic compartments, and $I^{syn}$ is the total synaptic current input to the neuron (see next section for details). IN neurons contained all intrinsic currents present in PY with the exception of the $I^{NaP}$. All intrinsic ionic currents ($I^j$) were modeled in a similar form:

$$I^j = g_j m^M h^N (V - E_j)$$

where $g_j$ is the maximal conductance, $m$ (activation) and $h$ (inactivation) are the gating variables, $V$ is the voltage of the corresponding compartment, and $E_j$ is the reversal potential of the ionic current. The gating variable dynamics are described as follows:

$$\frac{dx}{dt} = -\frac{x - x_\infty}{\tau_x},$$

$$\tau_x = \frac{(1/(\alpha_x + \beta_x))}{Q_T},$$

$$x_\infty = \frac{\alpha_x}{(\alpha_x + \beta_x)},$$

where $x = m$ or $h$, $\tau$ is the time constant, $Q_T$ is the temperature related term, $Q_T = Q^{((T-23)/10)} = 2.9529$, with $Q = 2.3$ and $T = 36$.

Thalamic neurons (TC and RE) were modeled as single compartment neurons with membrane potential dynamics mediated by the following equation:

$$C_m \frac{dV_D}{dt} = -I^{Na} - I^K - ACh_{gkl}I^{KL} - I^T - I^h - I^L - I^{syn},$$

where $I^{Na}$ is the fast Na$^+$ current, $I^K$ is the fast K$^+$ current, $I^{KL}$ is the K$^+$ leak current, $I^T$ is the low-threshold Ca$^{2+}$ current, $I^h$ is the hyperpolarization-activated mixed cation current, $I^L$ is the Cl$^-$ leak current, and $I^{syn}$ is the total synaptic current input to the neurons (see next section for details). The $I^h$ was only expressed in the TC neurons and not the RE neurons. The influence of histamine (HA) on $I^h$ was implemented as a shift in the activation curve by $HA_{gh}$ as described by:

$$m_\infty = \frac{1}{1 + exp\left(\frac{V + 75 + HA_{gh}}{5.5}\right)}.$$

A detailed description of the individual currents can be found in our previous studies (*Krishnan et al., 2016*; *Wei et al., 2018*).

## Synaptic currents and spike-timing dependent plasticity (STDP)

AMPA, NMDA, and GABA$_A$ synaptic current equations were described in detail in our previous studies (*Krishnan et al., 2016*; *Wei et al., 2018*). The effects of ACh on GABA$_A$ and AMPA synaptic currents in our model are described by the following equations:

$$I_{syn}^{GABA} = \gamma_{GABA_A} \, g_{syn} \, [O](V - E_{syn}),$$

$$I_{syn}^{AMPA} = ACh_{AMPA}\, g_{syn}\, [O]\big(V - E_{syn}\big),$$

where $g_{syn}$ is the maximal conductance at the synapse, $[O]$ is the fraction of open channels, and $E_{syn}$ is the channel reversal potential ($E_{GABA-A}$ = -70 mV, $E_{AMPA}$ = 0 mV, and $E_{NMDA}$ = 0 mv). Parameter $\gamma_{GABA_A}$ modulates the GABA synaptic currents for IN-PY, RE-RE, and RE-TC connections. For IN neurons $\gamma_{GABA_A}$ was 0.22 and 0.44 for awake and N3 sleep, respectively; $\gamma_{GABA_A}$ for RE was 0.6 and 1.2 for awake and N3 sleep. $ACh_{AMPA}$ defines the influence of ACh levels on AMPA synaptic currents for PY-PY, TC-PY, and TC-IN. $ACh_{AMPA}$ for PY was 0.133 and 0.4332 for awake and N3 sleep. $ACh_{AMPA}$ for TC is 0.6 and1.2 for awake and N3 sleep.

Spontaneous miniature excitatory post-synaptic potentials (EPSPs) and inhibitory post-synaptic potentials (IPSPs) were implemented for PY-PY, PY-IN, and IN-PY connections. The synaptic dynamics were similar to regular post-synaptic potentials (PSPs) described above and their arrival times were modeled by a Poisson process with time-dependent mean rate, with next release time $t_{release}$ given by:

$$t_{release} = (2/(1 + exp(-(t - t_0)/\upsilon)) - 1)/250\,,$$

where $t_0$ is the time of the last presynaptic spike. The maximal conductances for miniature PSPs were $g_{mini(PY-PY)}^{AMPA} = 0.03\,\mu S$, $g_{mini(PY-IN)}^{AMPA} = 0.02\,\mu S$, and $g_{mini(IN-PY)}^{GABA} = 0.02\,\mu S$. $\upsilon$ is the mini PSP frequency: $\upsilon_{mini(PY-PY)}^{AMPA} = 30$, $\upsilon_{mini(PY-IN)}^{AMPA} = 30$, and $\upsilon_{mini(IN-PY)}^{GABA} = 30$. Short-term depression of intracortical AMPA synapses was included. The maximal synaptic conductance was multiplied by a depression variable ($D \leq 1$), which represents the amount of available 'synaptic resources' as described in *Bazhenov et al., 2002*. This short-term depression was modeled as follows:

$$D = 1 - (1 - D_i(1-U))exp\left(-\frac{t - t_i}{\tau}\right)$$

where $D_i$ is the value of $D$ immediately before the $i_{th}$ event, $(t - t_i)$ is the time after the $i_{th}$ event, $U = 0.073$ is the fraction of synaptic resources used per action potential, and $\tau = 700ms$ is time constant of recovery of synaptic resources.

Potentiation and depression of synaptic weights between PY neurons were regulated by spike-timing dependent plasticity (STDP). The changes in synaptic strength ($g_{AMPA}$) and the amplitude of miniature EPSPs ($A_{mEPSP}$) have been described previously (*Wei et al., 2018*):

$$g_{AMPA} \leftarrow g_{AMPA} + g_{max}\, F(\Delta t),$$

$$A_{mEPSP} \leftarrow A_{mEPSP} + fA_{PY-PY}\, F(\Delta t),$$

where $g_{max}$ is the maximal conductance of $g_{AMPA}$, and f = 0.01 represents the slower change of STDP on $A_{mEPSP}$ as compared to $g_{AMPA}$. The STDP function F is dependent on the relative timing ($\Delta t$) of the pre- and post-synaptic spikes and is defined by:

$$F(\Delta t) = \begin{cases} A_+\, e^{-|\Delta t|/\tau_+}, & if\ \Delta t > 0 \\ -A_-\, e^{-|\Delta t|/\tau_-}, & if\ \Delta t < 0 \end{cases}$$

where $A_{+/-}$ set the maximum amplitude of synaptic change. $A_{+/-}$ = 0.002 and $\tau_{+/-}$ = 20 ms. $A_-$ was reduced to 0.001 during training to reflect the effects of changes in acetylcholine concentration during focused attention on synaptic depression during task learning observed experimentally (*Blokland, 1995*; *Shinoe et al., 2005*; *Sugisaki et al., 2016*).

## Sequence training and testing

Training and testing of memory sequences was performed similar to our previous study (*Wei et al., 2018*). Briefly, trained sequences were comprised of 5 groups of 10 sequential PY neurons. Each group of 10 were sequentially activated by a 10 ms DC pulse with 5 ms delay between subsequent group pulses. This activation scheme was applied every 1 s throughout the duration of the training period. Sequence 1 (S1) consisted of PY groups (in order of activation): A(200-209), B(210-219), C(220-229), D(230-239), E(240-249). Sequence 2 (S2) consisted of PY groups (in order of activation): W

(360-369), V(350-359), X(370-379), Y(380-389), Z(390-399) and can be referred as non-linear due to the non-spatially sequential activations of group W, V, and X. Sequence 1* (S1*) was trained over the same population of neurons trained on S1 but in the reverse activation order (i.e. E-D-C-B-A). During testing, the network was presented with only the activation of the first group of a given sequence (A for S1, W for S2, and E for S1*). Performance was measured based on the network's ability to recall/ complete the remainder of the sequence (i.e. A-B-C-D-E for S1) within a 350 ms time window. Similar to training, test activation pulses were applied every 1 s throughout the testing period. Training and testing of the sequences occurred sequentially as opposed to simultaneously as in our previous study (*Wei et al., 2018*).

## Data analysis

All analyses were performed with standard MatLab and Python functions. Data are presented as mean ± standard error of the mean (SEM) unless otherwise stated. For each experiment a total of 10 simulations with different random seeds were used for statistical analysis.

### Sequence performance measure

A detailed description of the performance measure used during testing can be found in *Wei et al., 2018* and the code is available in (https://github.com/o2gonzalez/sequencePerformanceAnalysis; copy archived at https://github.com/elifesciences-publications/ sequencePerformanceAnalysis; *González, 2020a*). Briefly, the performance of the network on recalling a given sequence following activation of the first group of that sequence (see *Methods and Materials*: *Sequence training and testing*) was measured by the percent of successful sequence recalls. We first detected all spikes within the predefined 350 ms time window for all 5 groups of neurons in a sequence. The firing rate of each group was then smoothed by convolving the average instantaneous firing rate of the group's 10 neurons with a Gaussian kernel with window size of 50 ms. We then sorted the peaks of the smoothed firing rates during the 350 ms window to determine the ordering of group activations. Next, we applied a string match (SM) method to determine the similarity between the detected sequences and an ideal sequence (ie. A-B-C-D-E for S1). SM was calculated using the following equation:

$$SM = 2*N - \sum_{i=1}^{N} |L(S_{test}, S_{sub}[i]) - i|,$$

where N is the sequence length of $S_{test}$, $S_{test}$ is the test sequence generated by the network during testing, $S_{sub}$ is a subset of the ideal sequence that only contains the same elements of $S_{test}$, and $L(S_{test}, S_{sub}[i])$ is the location of the element $S_{sub}[i]$ in sequence $S_{test}$. SM was then normalized by double the length of the ideal sequence. Finally, the performance was calculated as the percent of recalled sequences with SM≥Th, where Th is the selected threshold (here, Th = 0.8) indicating that the recalled sequence must be at least 80% similar to the ideal sequence to be counted as a successful recall as previously done in *Wei et al., 2018*.

### Sequence replay during N3 sleep

To find out whether a trained sequence is replayed in the trained region of the network during the Up state of a slow-wave in N3 sleep, we first identified the beginning and the end of each Up state by considering sorted spike times of neurons in each group. For each group, the time instances of consecutive spikes that occur within a 15 ms window were considered as candidate members of an Up state, where the window size was determined to decrease the chance of two spikes of the same neuron within the window. To eliminate spontaneous spiking activity of a group that satisfies the above condition but is not part of an Up state, we additionally required that the period between two upstate was at least 300 ms, which corresponds to a cortical Down state. The values for window durations reported above were identified to maximize the performance of the Up state search algorithm.

Once all Up states were determined, we defined the time instances when groups were active in each Up state. A group was defined as active if the number of neurons from the group that spikes during 15 ms exceeded the activation threshold, and the instance when the group is active was defined as the average over spike times of a subgroup of neurons with the size equals to the

activation threshold within the 15 ms window. In our study the activation threshold was selected to be half of a group size (i.e. five neurons). Using sorted time instances when groups are active, we counted the number of times a possible transition between arbitrary groups, and if all four transitions of a sequence were observed sequentially in the right order then we counted that as a replay of the sequence.

## Analysis of total sequence specific synaptic input

For every neuron from a group we computed the total synaptic weight 'from left' and 'to right', by considering the sum of all weights of synapses projecting to the neuron from neurons in preceding group, with respect to propagation of activity within a memory sequence if such a group exists, and the sum of all weights of synaptic connections from the neuron to the following group, if there is such a group. We omitted all synaptic connections within the group to which the neuron, for which the total synaptic weight is computed, belongs.

## Weight directionality index

To see how learning recruits neurons in encoding one of the competing sequences, we looked at the evolution of deviation from the center of unit square in a two dimensional subspace of total synaptic input from left and right neighboring neuronal groups. For this, we first found the total synaptic input from both sides, embedded it into a unit square, and computed Euclidean distance from the center of the square.

$$Weight\ directionality\ index = \sqrt{(li - 0.5)^2 + (ri - 0.5)^2},$$

where $li$ ($ri$) is the total synaptic input to a neuron from its left (right) neighboring neuronal group.

## Delay directionality index

To see whether neurons respond preferentially to one of the sequences, we evaluated signed (*Figure 8B*) and unsigned (*Figure 8C, D*) delay directionality indices, which are defined as follows. For each neuron, we found its response delays, $t_{S1}$ and $t_{S1*}$, after corresponding left and right neighboring neuronal groups were stimulated, respectively. Using these quantities, we computed the indices as

$$signed\ directionality\ index = \frac{\Delta t_{S1*} - \Delta t_{S1}}{\Delta t_{S1*} + \Delta t_{S1}},$$

$$unsigned\ directionality\ index = \frac{|\Delta t_{S1*} - \Delta t_{S1}|}{\Delta t_{S1*} + \Delta t_{S1}}.$$

## Acknowledgements

This work was supported by the Lifelong Learning Machines program from DARPA/MTO (HR0011-18-2-0021) and ONR (MURI: N00014-16-1-2829).

## Additional information

### Funding

| Funder | Grant reference number | Author |
|---|---|---|
| Defense Advanced Research Projects Agency | HR0011-18-2-0021 | Maxim Bazhenov |
| Office of Naval Research | MURI: N00014-16-1-2829 | Maxim Bazhenov |

The funders had no role in study design, data collection and interpretation, or the decision to submit the work for publication.

## Author contributions
Oscar C González, Yury Sokolov, Conceptualization, Data curation, Software, Formal analysis, Investigation, Visualization, Methodology, Writing - original draft, Writing - review and editing; Giri P Krishnan, Conceptualization, Software, Methodology, Writing - original draft, Writing - review and editing; Jean Erik Delanois, Data curation, Software, Formal analysis, Investigation, Visualization, Methodology, Writing - original draft, Writing - review and editing; Maxim Bazhenov, Conceptualization, Supervision, Funding acquisition, Writing - original draft, Project administration, Writing - review and editing

## Author ORCIDs
Oscar C González https://orcid.org/0000-0003-1302-1911
Yury Sokolov https://orcid.org/0000-0002-4590-3321
Giri P Krishnan http://orcid.org/0000-0002-3931-7633
Jean Erik Delanois https://orcid.org/0000-0002-8680-3239
Maxim Bazhenov https://orcid.org/0000-0002-1936-0570

## Decision letter and Author response
Decision letter https://doi.org/10.7554/eLife.51005.sa1
Author response https://doi.org/10.7554/eLife.51005.sa2

## Additional files

### Supplementary files
• Transparent reporting form

### Data availability
Computational models were used exclusively in this study. The model is fully described in the Methods section and code has been deposited to https://github.com/o2gonzalez/sequenceLearning-SleepCode (copy archived at https://github.com/elifesciences-publications/sequenceLearningSleepCode).

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
