## [Decision Letter]

**Acceptance summary:**

That sleep can protect and improve memories is well known, but many details remain unclear. This paper builds a detailed biological computational model to explore and demonstrate such effects.

**Decision letter after peer review:**

Thank you for submitting your article "Can sleep protect memories from catastrophic forgetting?" for consideration by *eLife*. Your article has been reviewed by three peer reviewers, one of whom is a member of our Board of Reviewing Editors, and the evaluation has been overseen by Laura Colgin as the Senior Editor. The following individual involved in review of your submission has agreed to reveal their identity: Francesco P Battaglia (Reviewer #1).

The reviewers have discussed the reviews with one another and the Reviewing Editor has drafted this decision to help you prepare a revised submission.

Summary:

Gonzalez and colleagues study a computational network model of sequence memory encoding and replay, in a thalamocortical network and find that sleep can untangle stored sequences.

Essential revisions:

As you will see there is a rather extensive list of comments and amendments requested. This is not typical for *eLife*, which aims to either reject or have small revisions. But we felt that if the issues are addressable, it would make a very interesting paper.

Reviewer #1:

Gonzalez and colleagues study a network model of memory encoding and replay, in a thalamocortical network.

The model, based on previous work from the same group, is a quite detailed rendition of neural dynamics (including Hodgkin-Huxley spike generation, and a host of other important conductances and neuromodulatory influences). This enables a fairly realistic depiction of the wake-sleep transition and of Up-Down states dynamics during NREM sleep.

The authors use several different protocols to implement training and replay of overlapping or non-overlapping sequences. Sleep appears to effective at stabilizing and orthogonalizing sequences, preventing catastrophic interference, as much, or in some cases more, than awake training.

The results are interesting in two ways. First, because they show, for the first time to my knowledge, that memory replay can happen and effectively supports consolidation in a realistic model of neural dynamics, and second because it shows some detail about how replay may support memory reorganization, orthogonalization, and protection from interference. Especially this second aspect could be improved in my view by expanding on some analyses, as detailed below:

– The Up/Down states are simulated in the network based on thalamocortical interactions. They are used in the analysis to "segment" neural activity. However, electrophysiological evidence (e.g. Johnson and McNaughton) shows that memory replay is strongest/most likely at the Down to Up state transition. Is this the case in these simulations as well?

– While many of the parameters have different values during wake and sleep, the values of the A_+_ constant in the STDP rule seem to be the same. If I understand correctly, A_-_ is different in training with respect to sleep (what happens at retrieval?). I was wondering how this affects the higher amount of orthogonalization seen with sleep replay with respect to interleaved awake training.

– Related to the previous point: is total synaptic strength decreasing, or increasing during NREM sleep? This would parallel ideas from e.g. Tononi and Cirelli or Grosmark and Buzsaki. Is greater synaptic depression during sleep a key ingredient for catastrophic interference protection?

– The effect of sleep seems to increase the amount of structure in each of the network blocks, with distinct groups of neurons forming for the sequences in the two directions (for example). This is studied mostly by looking at the synaptic level (asymmetry measure), and there, only by looking at an increase in the variability in the measure. That analysis could easily be extended in my view, by looking at neural activity measures (e.g. population vector correlation), more directly comparable with experimental results from neural ensemble recording. Is there evidence for the emergence of new within and across block structures? For example, to make the parallel with place cells (as done throughout the paper), is there evidence for the emergence, with training, of directional place cells from non-directional ones?

– The retrieval protocol used, in which all neurons in a "block" are strongly activated, may actually downplay the orthogonalizing power of the wake/sleep learning dynamics, because neurons that are assigned to, say, both the S1 and S1* sequences are primed. Is that the case? Wouldn't things look even "better" with a more gentle activation protocol?

– Figure 7B: after S1 training (second panel) the bulk of the neuron pairs shift, as expected, their asymmetry in the direction of S1, however, a smaller subset of neuron pairs go in the opposite direction, towards the K_01_ corner. Why is that?

Reviewer #2:

Here Gonzalez et al. implement a thalamocortical model to investigate how sleep rescues forgetting in a group of neurons competing for different memories. The study investigates sleep's role in protecting memories encoded in overlapping groups of neurons and provides important insights about the underlying mechanisms. The manuscript is well-written and easy to follow. They show that training a new sequence in a subpopulation that overlaps with a previous sequence leads to interference with the old sequence. Replay during Up states of sleep following this training however is able to reverse the damage caused by competing memories. This replay of competing memories during sleep creates subsets of strongly unidirectional synaptic connections for each learned sequence and some bidirectional connections that may act like network hubs. The approach is interesting and provides general insights into how spiking networks can create distinct but overlapping memories, and also makes some testable predictions.

The main issue, however, is that a key test of the effects of sleep replay on catastrophic interference was not performed: under realistic scenarios, sleep takes place after each sequence learning session separately and new memories are formed on top of pre-existing ones. E.g. the order in typical day-to-day experience is S1, sleep, S2, sleep, S1*, sleep, etc. This important scenario should be examined to test whether the results still hold. In the current models, the sequences seem to saturate specific synapses, which may create issues for future learning.

A second concern is that "replays" are inferred but not explicitly shown. The replay sequence was measured in terms of pairs of neurons active in a given Up state instead of how well the trained sequence is represented. The authors explain that Figure 6C is "suggesting" that two sequences replay simultaneously, but this should be more explicitly shown or better illustrated, as it is an interesting and important prediction of the model. There should also be discussion of how this observation is to be reconciled with the body of literature indicating distinct replay of isolated sequences. Perhaps previous methods based on template matching would fail to detect such overlapping replays? In particular, Wikenheiser and Redish (Hippocampus, 2013) indicate bidirectional replay during sleep, which appears inconsistent with the current results, at least in the hippocampus.

A third issue concerns how memories are selected for replay. In the model it seems that stronger synapses lead to increased replay. This would suggest that replay begets more replay, until synapses are fully saturated. The effects of multiple sleep sequences and/or longer sleep durations should be simulated to examine this scenario, and (as noted above) whether new learning can take place in synapses saturated by sleep. It is also important to note that studies investigating hippocampal replay (e.g. O'Neill et al., 2007; Giri et al., 2019) show decreasing replay of familiar sequences. Do the authors predict different patterns in the cortex? Please discuss.

The authors present and discuss an asymmetric expansion of place fields following sleep. However, this result is not consistent with Mehta et al., 1997 as the authors claim. Mehta in fact found that the backwards expansion of place fields was reset at the beginning of each track session (presumably as a result of overnight sleep), but that the place-fields expanded again over the course of a behavioral session. The findings of Mehta et al. are therefore more consistent with net downscaling of memory during sleep, rather than the model presented here. This discrepancy should be addressed.

Reviewer #3:

This paper examines the formation of sequence memories in a computational model of hippocampus equipped with STDP, and in particular in the role of sleep.

Sleep, as well as interleaved training, is shown to separate storage of forward and backward sequences.

How this relates to catastrophic forgetting is a bit tentative. For instance, I don't think the setup will allow storage of both ABCDE and ABCFG, but it would be nice to be proven wrong.

I wonder whether the story has a simple interpretation: Balanced STDP (A_+_ = A_-_, τ_+_ = τ_-_) by itself breaks symmetry and will not allow bidirectional connections between neurons. Therefore during sleep recurrent connections are cleared up. During learning however the symmetry is explicitly broken by hand by reducing the LTD term (subsection “Synaptic currents and spike-timing dependent plasticity (STDP)”).

I wonder if the mechanism thus requires an extraordinary amount of fine-tuning that could not expected in biology. On the other hand, during interleaved training, during which the symmetry is presumably also broken by hand, bi-directional connections are also removed. As a result I am not fully convinced my explanation is correct. In any case, I think it would be important to show that the mechanism does not require fine-tuning of the A_+_/A_-_ ratio.

The paper could have been presented better. The Materials and methods are particularly poor, as it mentions for instance histamine modulation and minis, however, they are mentioned just once and it remains completely unclear whether these ingredients matter. Now it might well be that this was extensively discussed in previous papers, but that is no excuse.

Similarly the dynamics of the synaptic input (subsection “Synaptic currents and spike-timing dependent plasticity (STDP)” is not discussed.

---

## [Author Response]

Reviewer #1:Gonzalez and colleagues study a network model of memory encoding and replay, in a thalamocortical network.[…] The results are interesting in two ways. First, because they show, for the first time to my knowledge, that memory replay can happen and effectively supports consolidation in a realistic model of neural dynamics, and second because it shows some detail about how replay may support memory reorganization, orthogonalization, and protection from interference. Especially this second aspect could be improved in my view by expanding on some analyses, as detailed below:

We would like to thank the reviewer for their interest in our study and for highlighting the importance of our work. We would like to add that by abstracting the basic mechanisms of sleep replay as demonstrated in this manuscript, we were able to construct sleep-like algorithms which can be applied to deep learning artificial neural networks to help overcome catastrophic forgetting and interference without the need to explicitly retrain old tasks (arXiv:1908.02240v1).

– The Up/Down states are simulated in the network based on thalamocortical interactions. They are used in the analysis to "segment" neural activity. However, electrophysiological evidence (e.g. Johnson and McNaughton) shows that memory replay is strongest/most likely at the Down to Up state transition. Is this the case in these simulations as well?

We would like to thank the reviewer for this question. Indeed, there is substantial evidence from the McNaughton lab and other groups that replay of the recently encoded memories tends to occur at the Down to Up transition.

To check what happens in the model, we located individual Up states and counted total number of partial (pair-wised) replays during the 1st vs. 2nd half of all Up states (Author response image 1). The majority of these transitions happened during the first half of an Up state. This predicts that the partial replays of the sequences during cortical Up state is strongest near the Down to Up transition but is not limited to that specific phase of an Up state.

**Author response image 1. respfig1:** Total number of partial sequence replays for sequence S1 (blue) and S1* (red) during the first half (left) and last half (right) of up states during sleep.

We would like to add that the evidence about replay being dominant near Down-Up state transition comes from *in vivo* studies in which multiple brain regions including the hippocampus and cortex are in continual communication during SWS. Indeed, it has been shown that hippocampal sharp-wave ripples (SWRs) tend to occur at the cortical Down to Up transition (Skelin et al., 2019). As such, the presence of hippocampal ripples, triggering replay, may be an important ingredient in the observation of the new memory replays at the Down to Up transition. In this study, we do not explicitly model the hippocampus or hippocampal input to the cortex. Instead we assumed that memory traces are already embedded to the cortical connectivity matrix either because of the earlier hippocampus-dependent consolidation or because these memories are hippocampus independent (as for procedural memory tasks) and we studied spontaneous cortical replay of these memories. Thus we predict that the cortical memory traces that are already in the cortex also tends to replay more during initial phase of Up state but replay continues through the entire Up state duration. This would further predict that hippocampus dependent replay of the memories that are not yet encoded in the cortex would occur earlier in Up state compare to the spontaneous cortical replay. We included a brief discussion of that topic to the revised text (see below).

In Results:

“Note, our model predicts that partial sequences (specifically spike doubles) of both memories can be replayed during the same Up state and not that both are replayed simultaneously (at the same exact time). […] This result is consistent with electrophysiological data suggesting that memory replay is strongest at the Down to Up state transition (Johnson et al., 2010).”

In Discussion:

“There are evidences that memory replay during SWS occurs predominantly near the Down to Up state transitions (Johnson et al., 2010). […] This predicts that hippocampus-dependent replay of new memories, that are not yet encoded in the cortex, may occur earlier in the Up state compared to the spontaneous replay of the old memory traces, which may occur later in the Up state.”

– While many of the parameters have different values during wake and sleep, the values of the A_+_ constant in the STDP rule seem to be the same. If I understand correctly, A_-_ is different in training with respect to sleep (what happens at retrieval?). I was wondering how this affects the higher amount of orthogonalization seen with sleep replay with respect to interleaved awake training.

Thank you for this comment. We should be clearer about how A_-_ was changed in the model during wake as compared to sleep. During the training phase, A_-_ was reduced to promote LTP, to simulate to some extend effects of local release of acetylcholine during attention learning (Blokland, 1995; Shinoe et al., 2005; Sugisaki et al., 2016). During testing/retrieval phases A_-_ was the same as A_+_. However, we want to say that during testing/retrieval reactivations of the memories resulted in very little to no change synaptic changes. (We also tested models where plasticity was completely frozen during testing/retrieval and obtained the same result.) Importantly, A_-_ and A_+_ were equal both during interleaved training and during sleep, to allow comparison between them. This is now emphasized in the revised text.

In Results:

“We set probabilistic connectivity (p = 0.6) between excitatory cortical neurons within a defined radius (RAMPA(PY-PY) = 20). Only cortical PY-PY connections were plastic and regulated by spike-timing dependent plasticity (STDP). […] STDP remained balanced during both sleep and interleaved training (except for few selected simulations where we tested effect of unbalancing STDP) to allow side by side comparisons.”

“It should be noted that though testing resulted in reactivation of memory traces, there was little change in synaptic weights during testing periods because of a relatively small number of pre/post spike events. (Simulations where STDP was explicitly turned off during all testing periods exhibited similar results to those presented here.)“

To show that the orthogonalization of the memory traces during sleep is independent of the specific balance of LTP/LTD (A_+_/A_-_), we have performed additional simulations. As shown in Figure 7—figure supplement 2, biasing the LTP/LTD ratio during sleep towards either LTD (Figure 7—figure supplement 2A) or LTP (Figure 7—figure supplement 2B) still results in the orthogonalization of the memory representations as demonstrated by the increase of sequence specific connections after sleep. In these figures, the scatter plots show the distribution of bidirectional synaptic connections in the trained region at baseline (left), after S1 training (middle left), after S1* training (middle right), and after sleep (right). (As in the main paper, for each bidirectional coupling between two neurons A<->B , we plotted the strength of A->B synapse on X axis and the strength of B->A synapse in Y axis.) The red lines on these plots depict the threshold used to identify neuronal pairs that are either strongly preferential for S1 (bottom right corner) or S1* (top left corner). The amount of synapses above this threshold are quantified in the bar plots below showing that sleep increases the density of the memory specific couplings between neurons regardless of the LTP/LTD ratio.

The vector field plots (Figure 7—figure supplement 2, middle panels) provide summaries of the average synaptic weight dynamics during training of either sequence (left and middle plots) and during sleep. It reveals convergence towards the corners which represent cell pairs being strongly enrolled either to sequence S1 or sequence S1*. It should be noted, because our model does not have homeostatic mechanisms to regulate “average” synaptic strength during sleep, the case of LTD biased sleep shows a net reduction of synaptic strengths (vector fields point towards the origin), while the LTP biased condition shows a net increase (vector fields pointing to top right corner). It is interesting that in the case of net reduction of synaptic weights, there is still strong orthogonalization of the weights. So, what was happening is that the bulk of synapses decreases their strength while fraction of synapses kept or even increased the strength and these synapses became memory specific. This observation may be in line with ideas from the Tononi and colleagues showing net reductions of synaptic weights during sleep (Tononi and Cirelli, 2014), however, more analysis of the model including additional homeostatic rules is need to make this conclusion from our simulations.

These results are discussed as follows.

In Results:

“In all previous simulations, LTP and LTD were balanced during sleep or interleaved training. […] This observation may be in line with ideas from Tononi and colleagues showing net reductions of synaptic weights during sleep (Tononi and Cirelli, 2014) however, more analysis of the model including additional homeostatic rules is need to make this conclusion based on model simulations.”

– Related to the previous point: is total synaptic strength decreasing, or increasing during NREM sleep? This would parallel ideas from e.g. Tononi and Cirelli or Grosmark and Buzsaki. Is greater synaptic depression during sleep a key ingredient for catastrophic interference protection?

Indeed, there has been much debate around the Sleep Homeostasis Hypothesis (SHY) put forward by Tononi and Cirelli (Tononi and Cirelli, 2014). In our current model, there is no explicit homeostatic mechanisms in place to bias towards synaptic depression and total synaptic strength change very little during simulations. As we mentioned in the previous comment, when plasticity was explicitly biased towards LTD during sleep (Figure 7—figure supplement 2A), sleep increased selected subset of synaptic weights which became memory specific while decreased the rest of synapses. This mechanism may aid in increasing the memory capacity of the network by only allowing the minimal number of connections required for the preservation of the sequences and resetting during sleep the other synapses towards the baseline. The network would then be able to use these synapses to encode new memories thus potentially facilitating continual learning without the consequence of retroactive interference. We included a short discussion of this point to the revised version.

In Discussion:

“The Sleep Homeostasis Hypothesis (Tononi and Cirelli, 2014) suggests that homeostatic mechanisms active during sleep should result in a net synaptic depression to renormalize synaptic weights and to stabilize network dynamics. […] The network would then be able to use these synapses to encode new memories thus potentially facilitating continual learning without the consequence of retroactive interference.”

– The effect of sleep seems to increase the amount of structure in each of the network blocks, with distinct groups of neurons forming for the sequences in the two directions (for example). This is studied mostly by looking at the synaptic level (asymmetry measure), and there, only by looking at an increase in the variability in the measure. That analysis could easily be extended in my view, by looking at neural activity measures (e.g. population vector correlation), more directly comparable with experimental results from neural ensemble recording. Is there evidence for the emergence of new within and across block structures? For example, to make the parallel with place cells (as done throughout the paper), is there evidence for the emergence, with training, of directional place cells from non-directional ones?

We have looked at activity-based directionality index at the very end of the manuscript (old Figure 8). For that we stimulated independently every single group – A, B, C, D, E – and we measured and compared the response delay for each neuron in groups B-D when its respective left vs. right neighboring groups were stimulated. This analysis is similar to what was done in Navratilova et al., 2012, where the difference of place cell responses at a specific locations on a liner track was calculated when a rat was approaching that location from one direction vs. the other. We used this delay-based measure instead of (spatial) population vector correlation since, due to the dense local connectivity in our model. Because of that, neurons within a group, even when do not encode specific memory explicitly, still would fire in response to elevated activity within that group, however, their response is delayed. As shown in Figure 8C, the directionality index based on neural activity shows a similar result as what was reported by (Navratilova et al., 2012) and it correlates with the asymmetry measured in the weight based directionality index (Figure 5E). In both measures, we see increases in the directionality or asymmetry after sleep (Figures 5E and 8C red bar).

– The retrieval protocol used, in which all neurons in a "block" are strongly activated, may actually downplay the orthogonalizing power of the wake/sleep learning dynamics, because neurons that are assigned to, say, both the S1 and S1* sequences are primed. Is that the case? Wouldn't things look even "better" with a more gentle activation protocol?

This is interesting point. In principle, if stimulation (e.g., group A) would be selective to activate only subset of neurons in A projecting alone the sequence in testing (e.g., sequence S), this should be sufficient to trigger sequence recall. Small difference between that and stimulating entire group A is that while other neurons in A may not project to the next group (group B), their activation could still increase overall spiking of the sequence specific neurons in A because of the intra group connectivity. Also, in many cases, nonspecific synapses between groups still preserve some (low) synaptic strength even after training or sleep, so activating these presynaptic neurons in A helps increasing excitability of the group B neurons. Thus, while the reviewer is correct that sequence specific stimulation is sufficient, activating more neurons at the beginning of the sequence would not have a negative impact on the sequence recall and may even facilitate initiation of the sequence. From practical perspective, to get selective activations would require online detection of neurons which have become preferential for one sequence or the other after sleep and selectively activate them during testing. Alternatively, we could detect these neurons offline, and then rerun the simulation and only stimulate the sequence specific neurons identified offline. Both of which are possible but require repeating many simulation. We would like to test this approach in our future work.

– Figure 7B: after S1 training (second panel) the bulk of the neuron pairs shift, as expected, their asymmetry in the direction of S1, however, a smaller subset of neuron pairs go in the opposite direction, towards the K_01_ corner. Why is that?

This is very good question. We checked and we found that a small population of recurrent/bidirectional synaptic weights that change their strength in direction opposite to what is expect from training are “within group” connections (i.e., a subset of synaptic connections formed between neurons within a group, e.g., within group B only). Because of the variability of the neuronal dynamics, when DC pulse is applied to a single group during training, there is some spike time variability of responses within a group. As such, connections may increase in one or another direction based on the exact timing of the spiking of neurons within that group. It is interesting to note that while these within group connections increase in the “opposite” direction, they do not impair the consolidation of the trained memory. This further supports our findings that the segregation of between group connections is main mechanism by which sleep aids in reversing the effects of catastrophic interference. To address this point we have added the following text to the Results section to provide an explanation of this phenomenon.

In Results:

“It should be noted that a small subset of synaptic weights increased in the direction of S1* during S1 training. Analysis of this small population of synaptic weights revealed that these connections were comprised solely of “within group” connections. It is interesting to note that these synapses did not impair the consolidation of the trained memory but instead helped to increase activity within each group regardless of which sequence was recalled.”

Reviewer #2:[…] The main issue, however, is that a key test of the effects of sleep replay on catastrophic interference was not performed: under realistic scenarios, sleep takes place after each sequence learning session separately and new memories are formed on top of pre-existing ones. E.g. the order in typical day-to-day experience is S1, sleep, S2, sleep, S1*, sleep, etc. This important scenario should be examined to test whether the results still hold. In the current models, the sequences seem to saturate specific synapses, which may create issues for future learning.

Thank you for this comment. We believe that our training paradigm represent two potential scenarios: 1) two memories are learned sequentially followed by sleep; 2) first sequence is trained and consolidated during sleep prior to the new sequence training followed by another sleep episode.

As the reviewer noticed, in the manuscript we focused on the first scenario, however, we explicitly tested both conditions and they show similar results. We now include a new supplementary figure in the revised manuscript (Figure 5—figure supplement 1) presenting “S1, sleep, S1*, sleep” scenario (note we excluded S2 training because this memory does not interfere with other memories). In this simulation, we first trained S1 as described in the main text. Following S1 training the network was put to sleep so that further consolidation of S1 occurred (Figure 5—figure supplement 1A). S1 testing after sleep showed better completion of S1 and increased performance (Figure 5—figure supplement 1B/C).

Training of the competing memory (S1*) caused damage to the completion of S1 and reduced S1 performance while increasing S1* performance and completion (Figure 5—figure supplement 1B/C). With a period of sleep following S1* training, we once again see the orthogonalization of the memory traces. Performances for both sequences show an improvement after the final sleep episode (Figure 5—figure supplement 1C). Increasing the amount of S1* training from 350s (Figure 5—figure supplement 1A/B/C) to 450s (Figure 5—figure supplement 1D) resulted in further reduction of S1 performance, but both memories still showed increases in performance after the final period of sleep (Figure 5—figure supplement 1D). Thus, the training paradigm “S1, sleep, S1*, sleep” shows qualitatively very similar results of synaptic weights orthogonalization to the “S1, S1*, sleep” paradigm presented in the main text.

These results are discussed as following.

In Results:

“As we mentioned previously, the training protocol we have focused on in this study was of two memories trained sequentially before sleep. […] Thus, the training paradigm “S1 -> sleep -> S1* -> sleep” shows qualitatively similar results to the “S1 -> S1* -> sleep” paradigm.”

A second concern is that "replays" are inferred but not explicitly shown. The replay sequence was measured in terms of pairs of neurons active in a given Up state instead of how well the trained sequence is represented. The authors explain that Figure 6C is "suggesting" that two sequences replay simultaneously, but this should be more explicitly shown or better illustrated, as it is an interesting and important prediction of the model. There should also be discussion of how this observation is to be reconciled with the body of literature indicating distinct replay of isolated sequences. Perhaps previous methods based on template matching would fail to detect such overlapping replays? In particular, Wikenheiser and Redish (Hippocampus 2013) indicate bidirectional replay during sleep, which appears inconsistent with the current results, at least in the hippocampus.

These are very important questions and we will try to address them as follows. By saying in the text that the two sequences are replayed simultaneously, we meant to say that both sequences can be replayed during the same up-state (we clarified this in the revised text).

In Results:

“Note, our model predicts that partial sequences (specifically spike doubles) of both memories can be replayed during the same Up state and not that both are replayed simultaneously (at the same exact time).”

As a proof, we showed in the paper that pairs of neurons belonging to different sequences are found to be spiking in sequence specific order reliably during the same Up state. Part of the reason why we limited our analysis by the neuronal pairs is because, as the reviewer suggested, we had hard time using template matching algorithms to classify strongly overlapping spiking patterns from two sequences in our model. We do not claim that longer sequence replays do not exist in the model and we can actually see examples of triplets or even longer replays in case of single sequence training (Author response image 2, see long sequences of ordered spiking in the trained area selected with red lines even for nonlinear sequence. Note, we reordered blocks on neurons in the trained region to match training pattern), but for two overlapping sequences we had difficulty to detect them reliably. We want to add that in our previous models (e.g., Wei et al., 2018) it was much easier to detect full sequences (e.g., Figure 3) as we had identical number of synapses per cell, so spiking was very reliable. In the new model presented in this paper, connections are probabilistic (see Author response image 2 for distribution of number of connections per cell) so different neurons show much higher jitter in spiking delays. Thus, we limited our prediction to stating that cortical replay may occur in the form of partial replays in subsets of the full sequence. That is to say during an up-state, rather than replaying the entire sequence A-B-C-D-E, we would more likely replay individual transitions (e.g. A-B; D-E; C-D) (see Author response image 2). In this way overlapping sequences would be able to coexist in the same population of neurons and replay transitions for both sequences throughout the same Up-state. Recent data (Ghandour et al., 2019) has shown evidence for cortical partial memory replay of both recent and older memories. We discuss these important points in the revised text.

In Discussion:

“Our model predicts the possibility of the partial sequence replays, i.e., when short snippets of a sequence are replayed independently, within the cortex. […] Indeed, recent data (Ghandour et al., 2019) have shown evidence for partial memory replay during NREM sleep (also see (Swanson et al., 2020)).”

**Author response image 2. respfig2:** Sequence replays during sleep. (**A**) Representative up state showing full sequence replays within the trained region of the network. (**B**) Histogram of the total in-degree of the neurons in our model. The x-axis is the in-degree or total number of synaptic inputs between excitatory neurons. The y-axis indicates the number of excitatory neurons which receive a given number of synaptic inputs. (**C**) Average number of single / partial transition replays across all up states for S1 ( blue) or S1* (red).

We also want to say that synaptic weights analysis, if available, gives a reliable measure of the results of replay. The critical role of replay detection *in vivo* is because this is the only practical way of evaluating how synaptic representation of memories become strengthened during sleep. In the model, though we do not explicitly show full sequence replay, we have access to the synaptic weight information of every synapse in the network. As such, we directly measured their dynamics to show how their strengths are modulated during sleep.

Here we will address the reviewer’s comment regarding bidirectional replay. Forwards and backwards replay has been observed in rat hippocampus for recent memories. Within the hippocampus, it is interesting to note that backwards replay is predominantly observed during a post-task awake resting period (Foster and Wilson, 2006). The usefulness or importance of backwards replay has yet to be established. However, it has been suggested that during post-task quite wakefulness, backwards replay may reflect an evaluation of immediately preceding events (Foster and Wilson, 2006). Additionally, most of the evidence for backwards replay come from hippocampus. The bidirectionality of memory specific replay in the cortex has yet to be established. We would like to note that in our model one can see evidence for some amount of reverse replay (Figure 5—figure supplement 1A, “After S1” training panel). In this figure, we show the distribution of recurrent / bidirectional weights within the trained region. After training of S1, most of the synapses strengthen for S1, that is they move towards the bottom right corner of the plot. There exist a small population of weights which increase in the opposite direction which would suggest some amount of reverse reactivation in our model. However, these changes were mainly limited within group synapses so we should be careful to claim this to be evidence of the backwards replay in the model. As the reviewer mentioned bidirectional replay was reported in the hippocampus only, which may employ specific mechanisms not available in the cortex. We mention this limitation of the model in the revised text.

In Discussion:

“The phenomena of backwards memory replay and decrease in number of memory replays over time have been observed in rat hippocampus for recent memories. […] The opposite, in fact, may be true regarding the amount of old memory relays in cortex.”

A third issue concerns how memories are selected for replay. In the model it seems that stronger synapses lead to increased replay. This would suggest that replay begets more replay, until synapses are fully saturated. The effects of multiple sleep sequences and/or longer sleep durations should be simulated to examine this scenario, and (as noted above) whether new learning can take place in synapses saturated by sleep. It is also important to note that studies investigating hippocampal replay (e.g. O'Neill et al., 2007; Giri et al., 2019) show decreasing replay of familiar sequences. Do the authors predict different patterns in the cortex? Please discuss.

Indeed, in hippocampal recordings replay of recent memories decays over time. This may reflect gradual transition of memories from being hippocampus dependent to becoming hippocampus independent. Indeed, recent experimental evidence suggests that hippocampal sharp-wave ripples (SWRs) induce persistent synaptic depression

within the hippocampus (presumably to make it ready for storing new memories) and silencing of SWRs impairs future learning (Norimoto et al., 2018). We are not familiar with evidence suggesting that cortical memory replays decreases with time. There are experimental data supporting the idea of “replay begets more replay”, as it has been shown that during periods of NREM sleep (particularly stage 3 slow-wave sleep) cortical replay of recently formed memories results in a tagging of synapses involved in consolidation of those memories by increases in their synaptic efficacy (Langille, 2019). As such, it is likely that these tagged synapses will be reactivated throughout sleep thereby resulting in more cortical replay during both NREM and REM sleep periods (Diekelmann and Born, 2010; Langille, 2019).

As mentioned in our response to an earlier comment by the reviewer, we have explicitly tested the training paradigm of training sequence S1 – sleep – training sequence S1* – sleep. As shown in Figure 5—figure supplement 1, new memory S1* still can be trained and this condition also results in memory orthogonalization and segregation of synaptic weights. This is due to the continuous replays that occur in the model during slow wave sleep. From that perspective, model predicts that the cortex may continuously replay / reactivate subsets of the old memories across individual Up-states throughout sleep in order to maintain existing memories and to properly integrate newly formed memories into the existing framework without the interference with other previously stored memories.

Saying that, we have to consider the limitations of the model that may be missing some slow time scale mechanisms that control replay in the cortex. One can reasonable argue that continuous replay of everything is not necessary or may be waste of resources (even though cortical neurons are active during slow-wave sleep anyway) and there are ideas that replay should be mainly focusing on the old memories that are in danger of being erased by new learning, so replay should select memories based on similarities to the new knowledge. One possible mechanism of that may involve increase of activity in neural circuits involved in new memory reactivation so it will trigger replay of the old memories sharing the same circuits. We can see signs of that in our model as the region of interference is generally more active during slow-wave sleep but we believe making specific predictions regarding this important question would go beyond the scope of our model that my not include the biophysical mechanisms necessary to account for this phenomenon. We included a brief discussion of this important question to revised paper.

In Discussion:

“Cortical replay of recently formed memories results in a tagging of synapses involved in consolidation of those memories by increasing their synaptic efficacy (Langille, 2019). These tagged synapses may likely be reactivated throughout sleep thereby resulting in more cortical replay during both NREM and REM sleep (Diekelmann and Born, 2010; Langille, 2019).”

The authors present and discuss an asymmetric expansion of place fields following sleep. However, this result is not consistent with Mehta et al., 1997 as the authors claim. Mehta in fact found that the backwards expansion of place fields was reset at the beginning of each track session (presumably as a result of overnight sleep), but that the place-fields expanded again over the course of a behavioral session. The findings of Mehta et al. are therefore more consistent with net downscaling of memory during sleep, rather than the model presented here. This discrepancy should be addressed.

We thank the reviewer for this important comment. Indeed, Mehta et al., 1997 stated that the observed backwards expansion of place fields was reset between track sessions. It is important to note that the resetting of the backwards expanded place fields between sessions was a phenomenon specific to the CA1. More recent work found that this resetting effect is diminished in the CA3 (Roth et al., 2012). These results suggest possibility that different synapses (i.e. CA3 vs. CA1) may have different plasticity time constants. Furthermore, the neocortex may have entirely different synaptic dynamic since its goal is long term storage as opposed to temporal indexing. There are evidences that with successive sleep periods, cortical memories become hippocampal independent (Lehmann and McNamara, 2011) and this could explain why the reset of the place fields was observed in Mehta et al., 1997. In our work, we are providing a prediction about cortical memory consolidation as it may relate to similar phenomenon of backwards expansion in the hippocampus. To our knowledge these experiments have not been done yet. In all, we present here a prediction that the cortical (particularly in associate cortex) representation of the sequence memories undergoes a similar form of backwards expansion observed in hippocampus. This form of backwards expansion, however, is not expected to be reset as it is in hippocampus.

To address the second part of this comment regarding global downscaling, we have conducted additional simulations showing that LTD biasing during sleep still allows sleep in our model to orthogonalize the interfering memories while down scale majority of synapses (which are presumable not critical for that memory storage) (Figure 7—figure supplement 2A). These data predicts that during slow wave sleep many synaptic weights undergo a net reduction of strength, while there are still subsets of synapses which increase strength and become strongly preferential for the learned sequences.

The following text has been added to the Discussion section in order to address these comments.

In Discussion:

“It is important to note that (Mehta et al., 1997) found that backwards expansion of the place fields was reset between sessions. […] Our study predicts that the cortical (such as in associate cortex) representations of the sequence memories undergo a similar form of backwards expansion as it was observed in CA1. This form of backwards expansion, however, persists and even increases after sleep.”

Reviewer #3:This paper examines the formation of sequence memories in a computational model of hippocampus equipped with STDP, and in particular in the role of sleep.Sleep, as well as interleaved training, is shown to separate storage of forward and backward sequences.How this relates to catastrophic forgetting is a bit tentative. For instance, I don't think the setup will allow storage of both ABCDE and ABCFG, but it would be nice to be proven wrong.

The reviewer is correct. If all the memory info is compressed to just these sequences, ABCDE and ABCFG, which have identical initial subsequences (ABC), there is no information to decide what the rest of the sequence (DE and FG) should be continued. Perhaps, the main objective in this case would be what types of additional cues are needed to store and retrieve partially identical sequences. If these cues become part of the sequence, then initial segments are not identical and memories can be separated. Example of that may be how we select a path from our office to the parking lot vs. library. Initial segment is the same and we execute it based on previous learning (turn left, go to elevator, get to the exit). The rest (where to go after we left the building) however is different and we dice based on the additional knowledge (going to parking lot vs. library) which in such example should be considered part of the initial memory trace. More specifically, we imagine that it would be impossible for a mouse or a human to follow a specific sequence (ABCDE vs. ABCFG) if they are only presented with the same initial part of the sequence (ABC). What would be necessary for this type of task would be the addition of another “letter” or cue to help guide the trajectory of the rest of the behavior / response. That is to say, the condition proposed by the reviewer would work if, e.g., the sequences were augmented by adding K or L letter to the beginning (KABCDE vs. LABCFG). In this way, the network would have additional information to help guide its activity once it reached the sequence bifurcation point. And this information could be used in the network having appropriate long-range connections. Our main objective in considering two opposite sequences was that such a training protocol should allow forgetting of the originally trained memory after a new memory is trained using the same neuronal and synaptic resources.

I wonder if the mechanism thus requires an extraordinary amount of fine-tuning that could not expected in biology. On the other hand, during interleaved training, during which the symmetry is presumably also broken by hand, bi-directional connections are also removed. As a result I am not fully convinced my explanation is correct. In any case, I think it would be important to show that the mechanism does not require fine-tuning of the A_+_/A_-_ ratio.

We would like to thank the reviewer for this important question and insightful comments. Our motivation to use LTD biased STDP during training was because during focused learning the local release of acetylcholine results in a biasing of STDP towards LTP (Blokland, 1995; Shinoe et al., 2005; Sugisaki et al., 2016). STDP in the model was symmetric during sleep. To show that asymmetric STDP would lead to similar weights dynamics in the model during sleep and to address the reviewer’s comment with regards to the mechanism presented here requiring “fine-tuning” of the A_+_/A_-_ ratio, we have performed the following simulations and discuss their results below.

As shown in Figure 7—figure supplement 2, biasing the LTP/LTD ratio during sleep towards either LTD (Figure 7—figure supplement 2A) or LTP (Figure 7—figure supplement 2B) still results in the orthogonalization of the memory representations as demonstrated by the increase of sequence specific connections after sleep. In these figures, the scatter plots show the distribution of bidirectional synaptic connections in the trained region at baseline (left), after S1 training (middle left), after S1* training (middle right), and after sleep (right). (As in the main paper, for each bidirectional coupling between two neurons A<->B , we plotted the strength of A->B synapse on X axis and the strength of B->A synapse in Y axis.) The red lines on these plots depict the threshold used to identify neuronal pairs that are either strongly preferential for S1 (bottom right corner) or S1* (top left corner). The amount of synapses above this threshold are quantified in the bar plots below showing that sleep increases the density of the memory specific couplings between neurons regardless of the LTP/LTD ratio.

The vector field plots (Figure 7—figure supplement 2, middle panels) provide summaries of the average synaptic weight dynamics during training of either sequence (left and middle plots) and during sleep. It reveals convergence towards the corners which represent cell pairs being strongly enrolled either to sequence S1 or sequence S1*. It should be noted, because our model does not have homeostatic mechanisms to regulate “average” synaptic strength during sleep, the case of LTD biased sleep shows a net reduction of synaptic strengths (vector fields point towards the origin), while the LTP biased condition shows a net increase (vector fields pointing to top right corner). It is interesting that in the case of net reduction of synaptic weights, there is still strong orthogonalization of the weights. So, what was happening is that the bulk of synapses decreases their strength while fraction of synapses kept or even increased the strength and these synapses became memory specific. This observation may be in line with ideas from the Tononi and colleagues showing net reductions of synaptic weights during sleep (Tononi and Cirelli, 2014), however, more analysis of the model including additional homeostatic rules is need to make this conclusion from our simulations.

These results are discussed as follows:

In Results:

“In all previous simulations, LTP and LTD were balanced during sleep or interleaved training. […] This observation may be in line with ideas from Tononi and colleagues showing net reductions of synaptic weights during sleep (Tononi and Cirelli, 2014) however, more analysis of the model including additional homeostatic rules is need to make this conclusion based on model simulations.”

In Discussion:

“The Sleep Homeostasis Hypothesis (Tononi and Cirelli, 2014) suggests that homeostatic mechanisms active during sleep should result in a net synaptic depression to renormalize synaptic weights and to stabilize network dynamics. […] The network would then be able to use these synapses to encode new memories thus potentially facilitating continual learning without the consequence of retroactive interference.”

The paper could have been presented better. The Materials and methods are particularly poor, as it mentions for instance histamine modulation and minis, however, they are mentioned just once and it remains completely unclear whether these ingredients matter. Now it might well be that this was extensively discussed in previous papers, but that is no excuse.Similarly the dynamics of the synaptic input (subsection “Synaptic currents and spike-timing dependent plasticity (STDP)” is not discussed.

Thank you for this comment. In order to address these comments we have extended our Materials and methods section to reflect more discussion of the model.